# Graph neural processes and their application to molecular functions

## Abstract

Neural processes (NPs) are models for meta-learning which output uncertainty estimates. So far, most studies of NPs have focused on low-dimensional datasets of highly-correlated tasks. While these homogeneous datasets are useful for benchmarking, they may not be representative of realistic transfer-learning. In particular, applications in scientific research may prove especially challenging due to the potential novelty of meta-testing tasks. Drug discovery is one such research area that is characterized by sparse datasets of many functions on a shared molecular space. In this paper, we study the application of graph NPs to drug discovery with DOCKSTRING, a diverse dataset of docking scores. Graph NPs show competitive performance in few-shot learning tasks relative to supervised learning baselines common in chemoinformatics, as well as alternative techniques for transfer learning and meta-learning. In order to increase meta-generalization to divergent test functions, we propose fine-tuning strategies that adapt the parameters of NPs. We find that adaptation can substantially increase NP's regression performance while maintaining good calibration of uncertainty estimates. Finally, we present a Bayesian optimization experiment which showcases the potential advantages of NPs over Gaussian Processes in molecular applications.

## 1 Introduction

A major difficulty in the application of machine learning (ML) to drug discovery is the scarcity of labeled data. Experimental assays are expensive and time-consuming, and data collection is biased towards certain bioactivities (e.g. protein targets deemed medically relevant or commercially profitable) or molecules (e.g. those that are easier to acquire or synthesize). As a result, chemoinformatic datasets are highly sparse and non-overlapping. In a typical pharmaceutical company's chemical library, it is estimated that less than 1% of all the compound-assay pairs have been measured (Irwin et al., 2020). Even more strikingly, the public database ChEMBL (Mendez et al., 2019) is roughly 0.01% complete (ChEMBL).

Meta-learning, or "learning to learn", is a machine-learning paradigm that attempts to achieve fast adaptation to novel tasks given a small number of labeled datapoints (Finn et al., 2017). The meta-learning setting may be appropriate in drug discovery (Nguyen et al., 2020), where there exist a large amount of bioactivity functions that have been measured historically. Examples are physicochemical properties, protein binding affinities, phenotypic assays or ADMET endpoints. Typically, each function comprises too few datapoints to train a large neural model, but collectively a large set of bioactivities may be useful to learn biases of molecular functions, as well as molecular representations. However, given the sheer diversity of molecular functions that are available, extra care should be taken to ensure that the biases learnt during meta-training are adequate for meta-testing. For example, functions related to physicochemical properties, which are intrinsic to molecules, may be very different from *in vivo* cell assays, which depend on the complex interplay between molecules and a biological system (Bender & Cortes-Ciriano, 2021).

In addition to data efficiency and learning from sparse datasets, another feature that is desirable in drug discovery applications is the ability to produce uncertainty estimates (Thomas et al., 2022). This is especially important in settings that involve molecular selection and subsequent experimental validation, such as Bayesian optimization (BO) or virtual screening (VS). Experimental measurements are expensive and time-consuming, so commiting to the wrong set molecules can be extremely

costly to the medicinal chemist (Valerie Jentzsch, 2023). Well-calibrated uncertainty estimates can help users restrict their selection to molecules with reasonably confident predictions, or balance exploration and exploitation by selecting some novel but uncertain molecules along with others more conservative and certain. Neural processes (NPs) (Garnelo et al., 2018a;b) are a family of models for probabilistic meta-learning that can estimate the uncertainty of each prediction.

The primary contributions of this work are:

1. Benchmarking NPs on fingerprints (FPs) and molecular graphs (MGs) against a variety of single-task, transfer-learning and meta-learning baselines using DOCKSTRING, a diverse dataset of docking scores.

2. Proposing a simple yet effective parameter-adaptation strategy during meta-testing that increases meta-generalization to divergent test functions.

3. Showing empirically that the random sampling of contexts and targets while meta-training NPs contributes to their good calibration of uncertainty estimates.

## 2 RELATED WORK

Our work builds on a growing body of literature about the application of meta-learning to drug discovery. Nguyen et al. (2020) was an early work reporting the benefits of model-agnostic meta-learning (MAML) for bioactivity classification. More recently, Chen et al. (2023) presented ADKF-IFT, a deep Gaussian process (GP) for meta-learning molecular datasets that was evaluated on FS-Mol using FP representations. Like NPs, deep GPs are neural models that produce uncertainty estimates. Applying NPs to molecular property prediction has been explored recently for the first time (Lee et al., 2022; Garcia-Ortegon et al., 2022; Chan et al., 2023), yielding promising results. These works did not compare FP and MG representations, and it is unclear which is more advantageous in the meta-learning setting. Chan et al. (2023) addressed function heterogeneity in bioactivity datasets by clustering similar assays, effectively denoising predictions from across-assay variability. This strategy could be complementary to ours, which is based on parameter adaptation.

In the drug discovery literature, transfer learning in the low-data setting is often approached from the perspective of imputation of sparse datasets (Walter et al., 2022; Luukkonen et al., 2023). Alchemite (Irwin et al., 2020) is a commercial model for imputation of bioactivities. It resembles NPs in that it obtains information about the test function from a set of context points. A related family of models is proteochemometric models, which predict affinity values for protein-ligand pairs (Wikberg et al., 2003; Bongers et al., 2019). However, they rely on explicit protein representations, which makes them less general.

Specific benchmarks and datasets have been introduced to evaluate meta-learning methods in drug discovery. FS-Mol is a popular dataset for few-shot learning (FSL) with realistic bioactivity values from ChEMBL (Stanley et al., 2021). However, it is geared primarily towards classification, whereas NPs are usually used in a regression setting, and it splits molecules randomly, which may lead to an overestimation of performance (Sheridan, 2013; Martin et al., 2017; Simm et al., 2021). DOCKSTRING is a dataset of docking scores of 260k ligands against 58 diverse proteins (García-Ortegón et al., 2022). It is complete, allowing flexible sampling of variably-sized subsets, and it splits molecules by scaffold, which minimizes the risk of data leakage from chemical analogues.

## 3 MOLECULAR GRAPH NEURAL PROCESSES (MG-NPS)

### 3.1 NEURAL PROCESSES (NPS)

Consider a meta-training dataset with observations of real-valued functions $f_1, \ldots, f_n$, $f_i : \mathcal{X} \to \mathbb{R}$. In this paper, $\mathcal{X}$ represents the space of chemically feasible molecules, and $x \in \mathcal{X}$ refers to a single molecule represented either as a fingerprint vector (FP) or as a molecular graph (MG). Each molecular function $f_i$ is observed at a set of $O_i$ input points $x_o^i \in \mathcal{X}^{O_i}$, with known labels $y_o^i = \left(y_{o,1}^i, \ldots, y_{o,O_i}^i\right)$, where $y_{o,j}^i = f_i(x_{o,j}^i)$. Additionally, consider a meta-test function $f$, observed at a small set of $C$ *context* points $(x_c, y_c) = ((x_{c,1}, y_{c,1}), \ldots, (x_{c,C}, y_{c,C}))$. Our goal is to predict the values $y_t$ of $f$ at a set of $T$ *target* locations $x_t \in \mathcal{X}^T$ as accurately and efficiently as

possible, using the example context points $(x_c, y_c)$ and the observations from the example functions $f_i, \ldots, f_n$.

A neural process (NP) is a parametric model for meta-learning that aims to describe the predictive distribution $p\left(y_t \mid x_c, y_c \,; x_t\right)$[1]. NPs assume conditional independence between the targets and a Gaussian predictive distribution:

$$q\left(y_t \mid x_c, y_c \,; x_t\right) = \prod_{j=1}^{T} \mathcal{N}\left(y_{t,j} \,;\, \mu_\theta\left(x_{t,j}\right), \sigma_\theta^2\left(x_{t,j}\right)\right).$$

In this paper we use two flavors of NPs: the conditional NP (CNP) (Garnelo et al., 2018a) and the latent NP (LNP) (Garnelo et al., 2018b). Their predictive mean $\mu_\theta(x)$ and variance $\sigma_\theta^2(x)$ for an input $x$ are generated in three steps. First, contexts $(x_{c,j}, y_{c,j})$ are mapped by an encoder network $h_\theta$ to a local datapoint representation $r_j$. Then, all context encodings $r_j$ are combined into a global function encoding $r$ through a commutative operation, usually the sum or the mean. Commutativity guarantees invariance to contexts' permutations. Finally, a decoder network $g_\theta$ maps the function encoding $r$ and the input location $x$ to the predictive mean and variance. In the CNP, the decoding step is deterministic. In the latent LNP, decoding involves sampling a latent random variable $z$ from the approximate posterior $\tilde{q}_\phi$, which is then fed as an input instead of $r$ to the decoder network $g_\theta$. A summary of the encoding and decoding process is provided in Appendix A.1.

The parameters of the CNP $\psi = \{\theta\}$ are trained by backpropagation from the predictive log-likelihood $\mathcal{L}_\psi(y_t \mid x_c, y_c \,; x_t) = \log q_\theta(y_t \mid x_c, y_c \,; x_t)$. During meta-training, each meta-train function $f_i$ is seen once every epoch, but not all observations $(x_o^i, y_o^i)$ are used at each iteration. Rather, the $O_i$ observations are randomly subsampled to create two disjoint sets: a context set $(x_c^i, y_c^i)$ and a target set $(x_t^i, y_t^i)$, with sizes $C_i$ and $T_i$ respectively, $C_i + T_i \leq O_i$. The predictive log-likelihood on the current targets is optimized given the current contexts. Therefore, the final objective is

$$\mathbb{E}\left[\frac{1}{n}\sum_{i=1}^{n} \mathcal{L}_\psi\left(y_t^i \mid x_c^i, y_c^i \,; x_t^i\right)\right], \tag{1}$$

where the expectation is with respect to the random sampling procedure. $C_i$ and $T_i$ can themselves be stochastic: in our experiments, we sample them uniformly from $[20, 150)$ at each iteration. We find that this randomization is key to uncertainty quantification, making the model robust to varying context and target sizes at test time. In Section 6.3 we investigate the influence of these hyperparameters on the generalization of NPs.

The parameters of the LNP $\psi = \{\theta, \phi\}$ are trained by backpropagation from a function $\mathcal{L}_\psi(y_t \mid x_c, y_c \,; x_t) = \log q_\theta(y_t \mid x_c, y_c \,; x_t) + \rho(\phi, x_o, y_0)$, where a regularization term $\rho(\phi, x_0, y_0)$ reduces the sensitivity of the encoder to any given sample (Appendix A.2). This regularization is motivated by a variational Bayesian argument (Garnelo et al., 2018b). Meta-training is performed by randomising the context and target sets, as outlined previously, yielding an objective of the form (1).

## 3.2 Effective epochs

Since only some observations of a function $f_i$ are sampled as contexts or targets every epoch, how often an individual observation is seen will depend on the sample size relative to the total number of observations for that function $O_i$, which for molecular datasets can vary widely (Appendix B). In order to homogenize training across varying sample sizes and observed sets, we introduce the concept of *effective epochs* $e_e$, which we define as the average number of times an observed datapoint is seen during training. This quantity is calculated as

$$e_e^i = e\,\frac{\bar{n}}{O_i},$$

where $e$ is the number of epochs and $\bar{n}$ is the average sample size (sample size could itself be random). In our experiments, the average amount of contexts and targets while training is the same in every experiment, so effective epochs refer to views of an observation both as context and target.

---

[1]We use semicolon notation to differentiate contexts and targets, e.g. $q\left(y_t \mid x_c, y_c \,; x_t\right)$ in a predictive density or $\mathcal{L}(y_t \mid x_c, y_c \,; x_t)$ in an objective function. This distinction will be helpful in later sections, where the context points could themselves be predicted, e.g. $q\left(y_c \mid x_c, y_c \,; x_c\right)$ or $\mathcal{L}(y_c \mid x_c, y_c \,; x_c)$

### 3.3 Molecular graph attentive encoder

In order to apply NPs to molecular graphs (MGs), we expand the encoder module with a graph neural network (GNN) that processes atom and bond features and implements message passing with attention. Our architecture is similar to AttentiveFP (Xiong et al., 2020) but we remove recurrent units (GRUs) (Chung et al., 2014) and change attention to a query-key-value (QKV) mechanism Vaswani et al. (2017) to speed up computation. See (Xiong et al., 2020) and Appendix C.3.2 for a more detailed explanation of the architecture.

## 4 The challenge of meta-generalizing to real-world functions

The challenge of meta-generalization (or, similarly, meta-overfitting or meta-memorization) has been suggested previously (Yin et al., 2020). However, many meta-learning benchmarks are limited to functions of low diversity, which may inflate the perceived robustness of meta-learning methods. As a toy example of the challenge to meta-generalize to divergent test functions, we consider the set of 1D sinusoid functions used in the MAML few-shot learning evaluation (Finn et al., 2017). We perform a simple experiment to show that even a slight function modification such as a small change in sinusoid frequency is enough to compromise meta-generalization, in both MAML and CNPs (Figure 1 and Appendix D for details). If a small change in a simple function class over a 1D input space can disrupt meta-generalization, mismatches between meta-training and meta-testing in the real world are likely to cause problems too.

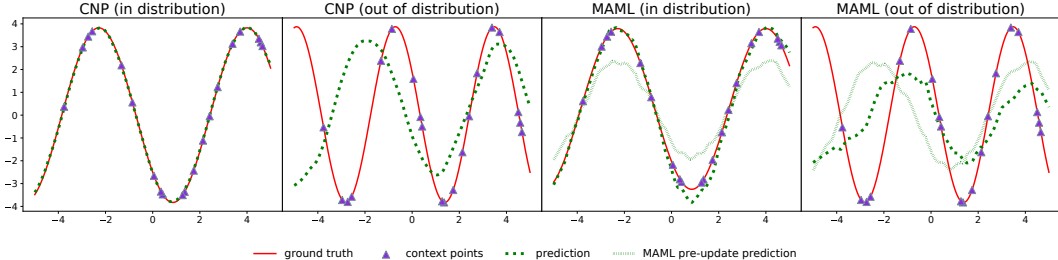

Figure 1: Meta-generalization experiment on 1D sinusoids. CNPs (left) and MAML (right) meta-trained on sinusoids with frequency $f = 1$ generalize to similar test functions. However, changing the frequency to $f = 1.5$ leads to catastrophic loss of generalization. Quantitative results are shown in Table D.1.

Drug discovery could potentially benefit greatly from meta-learning but such mismatches between meta-training and meta-testing may be unavoidable in scientific applications. First, in a research context novel functions of interest may not have known relatives available for meta-training. Furthermore, if the overlap of labeled datapoints is limited, it may not be obvious whether two functions are similar enough for meta-learning. This may occur in sparse bioactivity databases like ChEMBL. Finally, drug discovery encompasses a large variety of research questions and experimental techniques, which induces extreme function heterogeneity (Chan et al., 2023). While these difficulties complicate the application of meta-learning to drug discovery, transfering knowledge across very different, even seemingly unrelated functions is possible through parameter adaptation (Vinod et al., 2023). In this paper, we explore strategies to adapt the parameters of NPs to novel functions during meta-testing.

## 5 Parameter adaptation during meta-testing

When a NP is applied to a test function $f$ with contexts $(x_c, y_c)$, its predictions on the contexts themselves may be inaccurate. In particular, if a NP fails to meta-generalize, the predicted context density $q(y_c \mid x_c, y_c \,; x_c) = \prod_j^C \mathcal{N}(y_{c,j} \,; \mu_\theta(x_{c,j}), \sigma_\theta^2(x_{c,j}))$ may be inadequately low, even though the contexts are given as input. In this situation, the loss on the context predictions $\mathcal{L}_\psi(y_c \mid x_c, y_c \,; x_c)$ can be exploited to adapt the weights to the test function $f$, potentially improv-

ing meta-generalization. We studied two strategies for parameter adaptation by backpropagation during meta-testing: fine-tuning and a single step of gradient descent on a MAML-trained NP.

## 5.1 Fine-tuning

We fine-tuned NPs by mimicking the meta-training procedure on the test function $f$. To this end, at each epoch we split the test function's contexts $(x_c, y_c)$ into new contexts and new targets, as if they were the observations of a meta-train function. Then, we used the new contexts $(x_{c'}, y_{c'})$ and targets $(x_{t'}, y_{t'})$ to evaluate the NP's objective $\mathcal{L}_\psi(y_{t'} \mid x_{c'}, y_{c'}; x_{t'})$ and backpropagate as usual. In this way, fine-tuning during meta-testing resembled meta-training, but instead of iterating over many train functions every epoch, it focused on a single test function. As in meta-training, the new contexts and targets were disjoint, with sizes $C' + T' \leq C$.

We adapted the weights for 20 effective epochs, sampling $C' = T' = 20$ new contexts and targets every iteration. The number of epochs was adjusted based on the original context size $C$ to achieve the desired effective epochs. The only exception from this protocol was the few-shot learning experiment with 20 observations, where we used $C' = T' = 5$. To minimize the risk of overfitting, we fine-tuned the last layers and froze the rest. Specifically, we always adapted the last two layers of the decoder network $g_\theta$, and in LNPs we also adapted the last two layers of the encoder $g_\phi$ (see Appendices A.1 and C for architectural notation and details). The latter allowed us to optimize the regularization term of the LNP objective $\mathcal{L}_{\theta,\phi}$, which depends on the encoder (Appendix A.2).

## 5.2 Model-agnostic meta-learning (MAML)

Model-agnostic meta-learning (MAML) (Finn et al., 2017) is a meta-training approach to find model parameters that can be rapidly adapted to test functions in one or more steps of gradient-descent. At each meta-training iteration $t$, MAML simulates an adaptation experiment in two stages: the inner update (the simulated adaptation) and the outer update (the actual update of the model's parameters). First, during the inner update, it samples a set of support points from a test function, computes the loss of their predictions (inner loss), and takes a single step of gradient descent to adapt the model's parameters from $\theta_t$ to $\theta'_t$. Second, during the outer update, MAML samples another set of query points from the same function, computes the prediction loss on those points using $\theta'_t$ (outer loss), and backpropagates through the inner update to find new parameters $\theta_{t+1}$. Since the inner update involves a step of gradient descent, the outer update involves computing a gradient through a gradient, i.e. computing the Hessian of the parameters.

Similar to the support and query sets in MAML, meta-training NPs entails sampling two sets of observations: the contexts and the targets. The support set in MAML informs the prediction of the query set, analogous to how the context set in NPs informs the prediction of targets. We can meta-train NPs with MAML by computing the inner loss on the context points and the outer loss on the target points. At each iteration $t$, during the inner update a prediction on the contexts of a train function $f_i$ is made, with loss $\mathcal{L}_{\psi_t}(y^i_c \mid x^i_c, y^i_c; x^i_c)$. A single step of gradient descent is taken to adapt the parameters from $\psi_t$ to $\psi'_t$. During the outer update, a prediction on the targets is made using the adapted parameters $\psi'_t$, with loss $\mathcal{L}_{\psi'_t}(y^i_t \mid x^i_c, y^i_c; x^i_t)$. Finally, we backpropagate through the inner update to find new parameters $\psi_{t+1}$. Later, at meta-testing, we adapt NPs by taking a single iteration of gradient descent on the loss of the contexts of $f$, $\mathcal{L}_\psi(y_c \mid x_c, y_c; x_c)$.

Computing the Hessian of all MG-NP parameters was memory-prohibitive, so we only applied MAML to some layers, similar to Raghu et al. (2020). In particular, we applied it to the same layers adapted in the fine-tuning experiments. During meta-training, these layers underwent the inner and outer update cycle, while other layers experienced a single update per iteration, as usual. Later, during meta-testing, the MAML layers were adapted and the rest were frozen. In addition, backpropagating through gradients can lead to training instability, so we implemented modifications from Antoniou et al. (2019) to increase robustness.

# 6 EXPERIMENTS

## 6.1 FEW-SHOT LEARNING (FSL)

We evaluated few-shot learning (FSL) by MG-NPs on docking scores from the DOCKSTRING dataset and compared their performance to a variety of baselines, including single-task, transfer learning and meta-learning models. To recreate a low-data setting, we created a small training set by sampling a subset of 2.5k molecules from the DOCKSTRING training set, and a test set by sampling the same number from the DOCKSTRING test set (Appendix B). DOCKSTRING splits molecules by scaffold, which minimizes the risk of data leaking due to analogues (Sheridan, 2013; Martin et al., 2017; Simm et al., 2021). We trained all models on the 2.5k molecules from the training set. At test time, we evaluated meta-learning models with context points sampled randomly from the training set. We inspected a broad range of context points, from 20 to 1000, because the number of observations per function in bioactivity datasets may fluctuate considerably (Appendix B). Implementation details for all models are given in Appendix C.

Table 1: FSL of ESR, a protein with medium similarity to the meta-training set.

| | | ESR2 (medium correlation), $R^2 \cdot 100$ | | | | | |
|---|---|---|---|---|---|---|---|
| | Context set size | 20 | 50 | 100 | 200 | 500 | 1000 |
| Single task | Dummy regressor | -5.5 (1.9) | -3.5 (1.1) | -3.0 (1.1) | -3.5 (0.8) | -3.3 (0.4) | -3.1 (0.3) |
| | FP-RF | -5.3 (3.0) | 6.6 (2.0) | 11.2 (1.4) | 15.4 (1.4) | 21.0 (0.5) | 25.1 (0.2) |
| | FP-GP | **2.0 (1.0)** | **8.3 (1.3)** | **14.0 (1.3)** | **19.9 (0.9)** | **26.7 (0.4)** | **31.8 (0.4)** |
| | GNN | -10.7 (4.9) | -7.4 (3.6) | 1.8 (3.9) | 6.4 (3.9) | 14.8 (3.5) | 19.8 (4.2) |
| Transfer learning | GNN (random) | 5.7 (3.2) | 19.1 (2.1) | 10.8 (4.9) | 17.8 (2.5) | 24.3 (2.0) | 29.2 (1.6) |
| | GNN (fine-tuned) | **34.9 (1.5)** | **37.8 (0.8)** | **35.2 (1.1)** | **30.3 (1.2)** | **30.3 (1.0)** | **34.2 (0.6)** |
| Meta-learning | GNN (MAML) | 11.5 (11.1) | 12.7 (11.0) | 11.3 (10.5) | 10.8 (10.3) | 9.8 (10.4) | 11.1 (10.6) |
| | ADKF-IFT | -0.3 (1.4) | 8.1 (1.3) | 13.9 (0.9) | 18.0 (1.1) | 24.6 (0.7) | 29.4 (0.4) |
| | FP-CNP | 29.0 (0.8) | 26.0 (2.0) | 27.2 (1.3) | 28.8 (0.7) | 29.1 (0.6) | 29.7 (0.5) |
| | FP-LNP | -30.3 (1.2) | -30.3 (1.2) | -30.3 (1.2) | -30.3 (1.2) | -30.3 (1.2) | -30.3 (1.2) |
| | MG-CNP | **43.2 (1.0)** | **44.9 (0.7)** | **45.3 (0.6)** | **45.6 (0.5)** | **46.3 (0.5)** | **46.6 (0.6)** |
| | MG-LNP | 39.9 (1.3) | 40.0 (1.3) | 40.1 (1.2) | 40.3 (1.2) | 41.0 (1.0) | 41.5 (0.9) |
| NPs with parameter adaptation | FP-CNP (MAML) | 25.8 (2.6) | 24.5 (2.0) | 24.7 (1.4) | 26.7 (1.3) | 26.8 (1.0) | 27.9 (0.7) |
| | FP-CNP (fine-tuned) | 30.4 (0.8) | 29.5 (1.2) | 29.9 (0.9) | 31.4 (0.7) | 33.3 (0.3) | 34.0 (0.4) |
| | FP-LNP (MAML) | 6.8 (5.4) | 8.0 (4.6) | 8.2 (4.2) | 6.9 (4.3) | 7.4 (4.2) | 7.3 (4.2) |
| | FP-LNP (fine-tuned) | 29.6 (0.6) | 30.6 (0.6) | 29.8 (0.5) | 29.0 (0.8) | 30.4 (0.6) | 29.8 (0.7) |
| | MG-CNP (MAML) | 44.0 (0.9) | 44.9 (0.9) | 44.8 (1.2) | 45.9 (0.5) | 46.7 (0.5) | 47.0 (0.4) |
| | MG-CNP (fine-tuned) | **45.0 (0.9)** | **45.3 (0.8)** | **46.9 (0.8)** | **49.7 (0.4)** | **51.3 (0.4)** | **52.5 (0.4)** |
| | MG-LNP (MAML) | 38.4 (1.3) | 39.5 (0.9) | 39.9 (0.9) | 39.9 (0.9) | 40.5 (0.8) | 41.1 (0.7) |
| | MG-LNP (fine-tuned) | 43.3 (0.9) | 42.8 (0.9) | 46.2 (0.6) | 47.3 (0.7) | 49.5 (0.4) | 50.4 (0.6) |

| | | ESR2 (medium correlation), NLPD | | | | | |
|---|---|---|---|---|---|---|---|
| Single task | FP-GP | **0.52 (0.01)** | **0.40 (0.01)** | **0.35 (0.01)** | **0.31 (0.00)** | **0.26 (0.00)** | **0.22 (0.00)** |
| Meta-learning | ADKF-IFT | **0.41 (0.01)** | **0.36 (0.01)** | **0.32 (0.00)** | **0.29 (0.01)** | **0.25 (0.00)** | **0.21 (0.00)** |
| | FP-CNP | 1.69 (0.06) | 1.86 (0.10) | 1.76 (0.07) | 1.65 (0.04) | 1.63 (0.04) | 1.60 (0.04) |
| | FP-LNP | 1.52 (0.01) | 1.52 (0.01) | 1.52 (0.01) | 1.52 (0.01) | 1.52 (0.01) | 1.52 (0.01) |
| | MG-CNP | 1.04 (0.01) | 1.00 (0.01) | 0.98 (0.01) | 0.98 (0.01) | 0.97 (0.01) | 0.97 (0.01) |
| | MG-LNP | 1.08 (0.01) | 1.07 (0.01) | 1.07 (0.01) | 1.07 (0.01) | 1.06 (0.01) | 1.05 (0.01) |
| NPs with parameter adaptation | FP-CNP (MAML) | 1.69 (0.10) | 1.77 (0.13) | 1.70 (0.10) | 1.60 (0.05) | 1.57 (0.03) | 1.53 (0.02) |
| | FP-CNP (fine-tuned) | 1.18 (0.01) | 1.20 (0.02) | 1.16 (0.01) | 1.16 (0.01) | 1.17 (0.01) | 1.21 (0.01) |
| | FP-LNP (MAML) | 1.32 (0.04) | 1.30 (0.03) | 1.30 (0.03) | 1.31 (0.03) | 1.31 (0.03) | 1.31 (0.03) |
| | FP-LNP (fine-tuned) | 1.21 (0.02) | 1.15 (0.00) | 1.20 (0.01) | 1.22 (0.01) | 1.27 (0.01) | 1.34 (0.01) |
| | MG-CNP (MAML) | 1.03 (0.01) | 1.00 (0.01) | 0.99 (0.01) | 0.98 (0.01) | 0.97 (0.01) | 0.96 (0.01) |
| | MG-CNP (fine-tuned) | **0.98 (0.01)** | **0.95 (0.01)** | **0.94 (0.01)** | **0.91 (0.01)** | **0.89 (0.01)** | **0.87 (0.01)** |
| | MG-LNP (MAML) | 1.05 (0.01) | 1.04 (0.01) | 1.04 (0.01) | 1.04 (0.01) | 1.03 (0.00) | 1.02 (0.01) |
| | MG-LNP (fine-tuned) | 1.04 (0.02) | 1.01 (0.01) | 0.96 (0.01) | 0.95 (0.01) | 0.92 (0.01) | 0.91 (0.01) |

Table 1 shows ESR2, a protein whose docking scores have medium correlation to proteins in the meta-train set. The lower the similarity of a function to the meta-train set, the more challenging for meta-learning models, and the greater the potential benefit of parameter adaptation. Other targets with high correlation (PARP1) and low correlation (PGR) are shown in Appendix E. When the mismatch between meta-training and meta-testing is low (as for PARP1), all meta-learning methods outperform every single-task model in the low-data regime, as expected. However, when the

mismatch grows (ESR2 and PGR), the biases learnt during meta-training could become detrimental to prediction, leading to worse performance by meta-learning methods than even a dummy regressor in some cases. Parameter adaptation can greatly improve predictions, especially when there is a mismatch. In general, the best adaptation method appears to be fune-tuning, whereas MAML suffers from training instability, yielding large error bars, even after implementing modifications to increase robustness suggested by (Antoniou et al., 2019). In general, CNPs perform better than LNPs, although LNPs can achieve better negative log predictive density (NLPD) when the mismatch is low. Regarding molecular representations, MG-NPs perform significantly better than FP-NPs. The MG-CNP ranks consistently as the best model in terms of the coefficient of determination $R^2$ by a large margin. However, ADKF-IFT (Chen et al., 2023), a GP-based model with meta-learnt representations, shows the best calibration according to the NLPD. We will examine the MG-CNP in the following sections, and will compare the MG-CNP and ADKF-IFT in a BO experiment in Section 6.4.

## 6.2 CALIBRATION OF UNCERTAINTY ESTIMATES

NP predictions consist of a mean and variance value for each target point. Predictive variances can be viewed as an estimation of the confidence that the NP places on its own predictions. A model is well calibrated if, on average, lower predictive variances correlate with lower prediction errors. Figure 2 compares the calibration of MG-NPs to other models from Section 6.1. We selected 200 context datapoints at random (using the same data split as before, see Appendix B) and ranked the 2.5k target datapoints by predicted variance from most confident (lowest predicted variance) to most uncertain (highest predicted variance). Then, we partitioned them into 100 groups of 25 datapoints, which we call confidence percentiles; lower percentiles represent higher confidence. We computed the average mean square error (MSE) and the average predicted variance within each percentile.

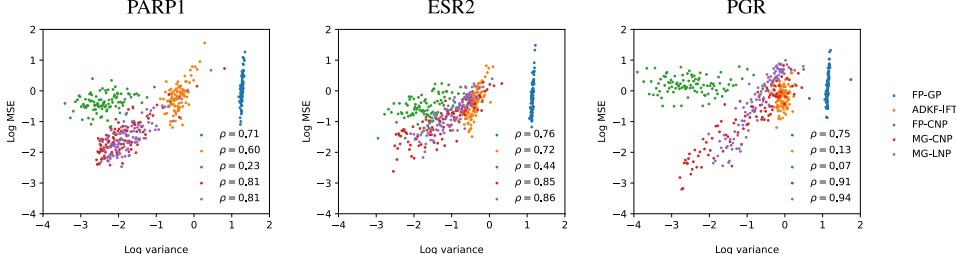

Figure 2: Calibration of uncertainty estimates from models in Section 6.1. Each point in the scatterplot indicates the log mean MSE and log mean predicted variance in each confidence percentile (percentiles of target datapoints ranked by predicted variance). The legend in each plot shows the Pearson correlation between the log MSE and the log variance for each model. The MG-CNP achieved the best correlation on the meta-test functions PARP1, ESR2 and PGR.

The highest correlation between the predicted variance and the prediction error was achieved by the MG-CNP, closely followed by the MG-LNP. Figure 3 examines the impact of fine-tuning these models on their uncertainty estimates, as well as calibration across different context sizes. On the left columns, the scatter plots show the log mean MSE and log mean variance of an unmodified NP (i.e. without parameter adaptation, in red) and a fine-tuned NP (in blue), using 200 random context points. The Pearson correlation between predicted variance and prediction error remains high in the fine-tuned NP ($\rho > 0.7$ in all cases), suggesting good calibration. On the right, the heatmaps show the MSE of the unmodified and fine-tuned models, using context sets ranging from 20 to 1000 datapoints. The row dimension indicates context set size and the column dimension shows test datapoints grouped by confidence percentile (the first column includes the 1st to the 10th percentile, the second the 11th to the 20th, etc). In each cell, the black and white scale represents the MSE of the unmodified model. As before, we observe that MG-NPs are well calibrated, with higher errors in the higher percentiles[2]. The red and blue scale indicates the difference between the

---

[2]Increasing the context set size (i.e. moving from top to bottom in the heatmaps of Figure 3) also improved performance and decreased MSE, as shown in Table 1 of the previous section and in Appendix E. However, the difference was small compared to the difference across confidence percentiles, so the latter dominates the black and white color scale.

MSE of the fine-tuned and the unmodified models. Blue cells signify that the fine-tuned model beat the unmodified one in that context size and confidence pair. In general, most fine-tuning gains came from the most erroneous unmodified predictions. This is to be expected: regions of the input space where predictions have higher loss will provide a larger signal for parameter adaptation.

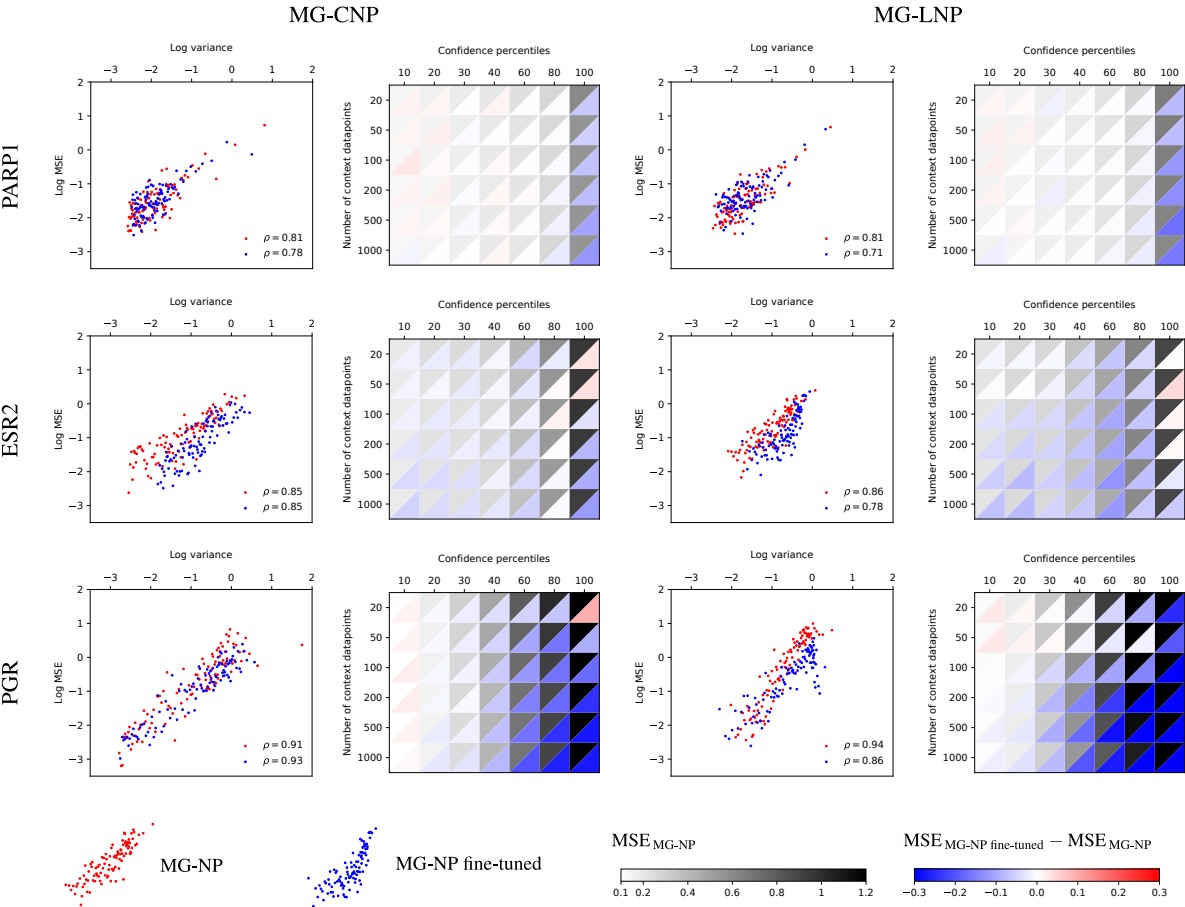

Figure 3: Impact of fine-tuning on uncertainty estimates. Scatterplots on the left show the log mean MSE and log mean predicted variance in each confidence percentile (percentiles of target datapoints ranked by predicted variance). The right heatmaps depict the absolute MSE of MG-NPs (black and white scale; black means higher error) and the MSE difference between the fine-tuned and the unmodified models (blue and red scale; blue indicates that fine-tuned is better).

## 6.3 RANDOMIZATION OF CONTEXT AND TARGET SETS PROTECTS NPS FROM OVERFITTING

The MG-CNP, a model with millions of parameters, exhibited good uncertainty calibration in meta-test functions despite having been trained with a maximum likelihood objective in as few as 2.5k molecules. Even though the CNP objective lacks a regularization term like that of the LNP, we did not find evidence that the model overfits, in the sense of underestimating posterior uncertainty. We hypothesized that this may be due to the randomization of context and target points during meta-training. The effect of this process may be two-fold: first, random sampling induces a combinatorially large number of unique views of each function, which may resemble an augmentation of the function set; and second, by using only a subset of all observations at each epoch, the number of effective epochs is kept low, hence reducing the risk of overfitting to any single datapoint.

In previous experiments, we trained MG-NPs using between 0.8% and 6% of observations as contexts and targets at each iteration. To investigate if random subsampling during meta-training protects NPs from overfitting, we trained a collection of MG-CNPs with the same architecture and on the same dataset as before, but using increasing fractions of observations as contexts and targets. As

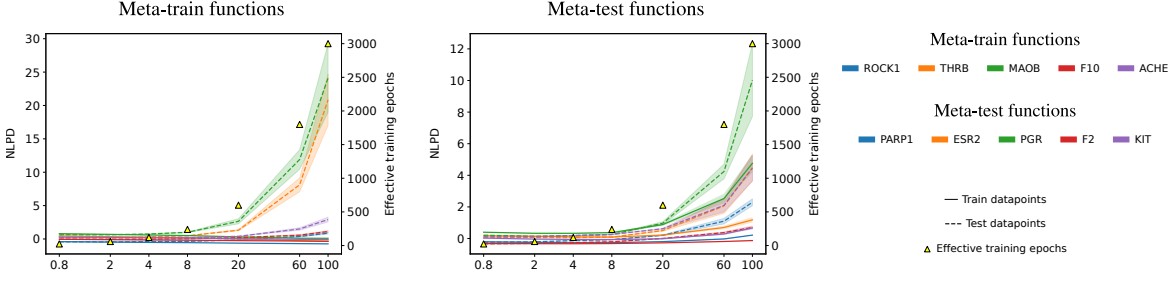

Figure 4: Random sampling of observations during meta-training protects from overfitting. Increasing the percentage of points sampled as contexts or targets leads to less unique function views and to more effective epochs. This causes overfitting, with memorization of the labels from the train points of the train functions (left, solid lines) and a degradation of performance on the test points of the train functions (left, dashed line) and all points of the test functions (right, solid and dashed lines).

the fraction of points sampled growed, MG-CNPs overfit to the training molecules of the train functions, both in terms of NLPD (Figure 4) and $R^2$ (appendix Figure F.1). In addition, the calibration of the uncertainty estimates deteriorated substantially (appendix Figure F.2). This suggests that, in order to achieve adequate meta-generalization and calibration, it is critical to tune the size of the context and target sets.

## 6.4 BAYESIAN OPTIMIZATION (BO)

We benchmarked the MG-CNP, ADKF-IFT and a GP on binary fingerprints in a sequential learning experiment using two objective functions from DOCKSTRING optimization benchmark, selective JAK2 and druglike F2. A description of the objectives is provided in Appendix G. For this experiment, we created a library of 60k molecular candidates by sampling 30k from the DOCKSTRING train set and 30k from the test set. We selected molecules from the library in batches of 5 at a time, up to a total budget of 1000, using a lower confidence bound (LCB) or greedy acquisition function (the latter selects molecules according to the best predicted mean). For meta-training, we created an augmented dataset of random combinations of transformed docking scores and QED (quantitative estimate of drug-likeness; see Appendix B for details). To avoid data leakage across functions, we excluded from the augmentation procedure the proteins that participated in the objectives, i.e. F2, JAK2 and LCK.

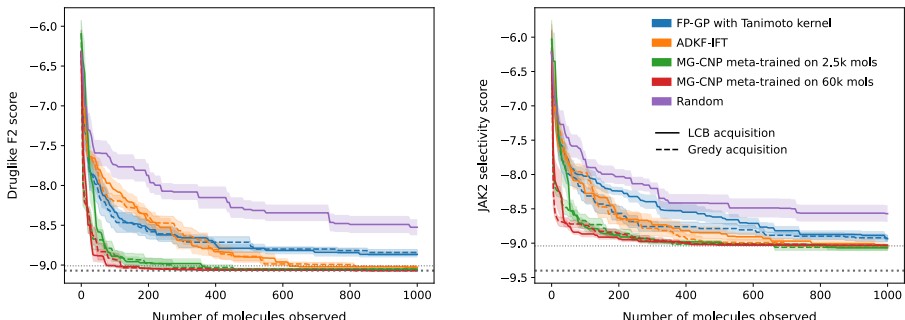

Figure 5: Bayesian optimization of druglike F2 (left) and selective JAK2 (right). MG-CNPs always reaches either the best or a near-best molecule outperforming ADKF-IFT and GPs with a Tanimoto kernel on binary FPs. Horizontal dotted lines indicate the best and second-best molecules in the library of 60k compounds.

We compared two MG-CNPs, one meta-trained on 2.5k molecules (Figure 5, orange) and another meta-trained on all 60k molecules from the library (green). The two MG-CNPs displayed similar BO trajectories, suggesting that NPs could be effective for molecular optimization even in the low-data setting. In druglike F2, they often reached the best molecule in the library or another with very

similar score (Figure 5, left). In selective JAK2, they always reached the second-best molecule in the library (Figure 5, right). As baselines, we compared to a GP with a Tanimoto kernel on binary FPs (blue), and to random selection (red). The GP performs significantly better than random but the MG-CNP always finds better molecules than the GP. Interestingly, LCB acquisition (solid line) did not do better than greedy acquisition (dashed line), neither in MG-CNPs or in the GP. While LCB with a well-calibrated model should perform better on expectation (i.e. given an infinite budget and choosing from a sufficiently large library of molecules), it may not always be beneficial in every data regime.

## 7 DISCUSSION

Drug discovery poses a unique challenge to meta-learning algorithms. Our large-scale benchmark demonstrates that NPs have competitive performance against baselines and enable transfer learning in realistic tasks. In addition, we propose fine-tuning strategies which address one of the main shortcomings of NPs — the inflexible way in which training points are used in meta-testing — showing that they significantly improve meta-generalization.

NPs are particularly attractive because they perform uncertainty quantification. We find that, despite the difference in their objective, CNPs and LNPs both provide well-calibrated uncertainty estimates. In addition, we show that randomization of context and target sets, and an adequate tuning of their size, are critical to calibration. Furthermore, our proposed finetuning in meta-test tasks does not degrade the quality of uncertainty estimates. Finally, we demonstrate the use of meta-learned MG-CNPs in a Bayesian Optimization experiment. This experiment illustrates the computational scalability and statistical efficiency of this class of algorithms.

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

# A NEURAL PROCESSES (NPS)

## A.1 PARAMETERIZATION OF CNP AND LNP

NPs assume conditional independence between the targets and a Gaussian predictive distribution with mean $\mu_\theta(x)$ and variance $\sigma_\theta^2(x)$:

$$q\left(y_t \mid x_c, y_c \,; x_t\right) = \prod_{j=1}^{T} \mathcal{N}\left(y_{t,j} \,;\, \mu_\theta\left(x_{t,j}\right), \sigma_\theta^2\left(x_{t,j}\right)\right).$$

The conditional NP (CNP) (Garnelo et al., 2018a) and the latent NP (LNP) (Garnelo et al., 2018b) parameterize the predictive mean and variance of an input $x$ in three steps:

- *Datapoint encoder*: a neural network $h_\theta$ maps each context point $(x_{c,j}, y_{c,j})$ to a local datapoint representation $r_j$.

$$r_j = h_\theta\left(x_{c,j}, y_{c,j}\right)$$

- *Function encoder*: all context encodings $r_j$ are combined into a global context encoding $r$ through a commutative operation $\oplus$, usually the sum or the mean. Commutativity of the function encoder guarantees invariance to permutations of the context set.

$$r = r_1 \oplus \cdots \oplus r_C$$

- *Decoder*: a neural network $g_\theta$ maps the function encoding $r$ and the input location $x$ to the predictive mean and variance. In the CNP, the decoding process is deterministic:

$$\left(\mu_\theta(x), \sigma_\theta^2(x)\right) = g_\theta\left(r, x\right).$$

  In the LNP, the decoding process involves sampling a latent variable. Its approximate posterior $q_\phi$ is parameterized as a Gaussian with mean and variance given by a neural network $g_\phi$.

$$z \sim q_\phi\left(z \mid x_c, y_c\right) := \mathcal{N}\left(\mu_\phi, \sigma_\phi^2\right), \quad \left(\mu_\phi, \sigma_\phi^2\right) = g_\phi(r)$$

  The final decoding stage continues like the CNP, but using $z$ instead of $r$:
$$\left(\mu_\theta(x), \sigma_\theta^2(x)\right) = g_\theta\left(z, x\right).$$

## A.2 OBJECTIVE FUNCTIONS OF NEURAL PROCESSES (NPS)

The CNP objective $\mathcal{L}_\theta$ is the conditional log likelihood of the targets given the contexts (Garnelo et al., 2018a).

$$\mathcal{L}_\theta\left(y_t \mid x_c, y_c \,; x_t\right) = \log q_\theta\left(y_t \mid x_c, y_c \,; x_t\right)$$

The LNP objective $\mathcal{L}_{\theta,\phi}$ consists of a reconstruction and a regularization term (Garnelo et al., 2018b):

$$\mathcal{L}_{\theta,\phi}\left(y_t \mid x_c, y_c \,; x_t\right) = \mathbb{E}_{q_\phi(z \mid x_d, y_d)}\left[\log p_\theta\left(y_t \mid z, x_t\right)\right] - D_{\mathrm{KL}}\left(q_\phi(z \mid x_d, y_d) \,\|\, q_\phi(z \mid x_c, y_c)\right),$$

where $x_d = (x_c, x_t)$, $y_d = (x_c, y_t)$ and $D_{\mathrm{KL}}$ is the KL divergence. It is an approximation of an evidence lower bound (ELBO) to the conditional log marginal likelihood:

$$\log p_\theta(y_t \mid x_d, y_c) \geq \mathbb{E}_{q_\phi(z \mid x_d, y_d)}\left[\log p_\theta(y_t \mid z, x_t)\right] - D_{\mathrm{KL}}\left(q_\phi(z \mid x_d, y_d) \,\|\, p_\theta(z \mid x_c, y_c)\right).$$

The ELBO is intractable due to the posterior $p_\theta(z \mid x_c, y_c)$ in the KL term. The LNP objective is derived by approximating the posterior with $q_\phi(z \mid x_c, y_c)$. This way the objective becomes tractable but $\mathcal{L}_{\theta,\phi}$ is no longer an analytical lower bound.

## B DATA

### B.1 DOCKSTRING SUBSETS USED IN META-LEARNING EXPERIMENTS

DOCKSTRING contains docking scores for 58 targets and approximately 260k molecules. Of these molecules, roughly 220k belong to the training set and the rest are part of the test set. The train and test sets are split by chemical scaffolds to minimize the risk of data leakage from chemical analogues. We respected the original split by sampling 2.5k compounds from the training set and 2.5k from the test set. Those were our training and test molecules in the few-shot learning (FSL) experiments.

Regarding the function split, we took advantage of the design of the the DOCKSTRING regression benchmark. This benchmark selects 5 diverse proteins, some of which have scores that are relatively easy to predict, and some of which are harder. We kept these 5 proteins for meta-testing and used the other 53 for meta-training.

Our FSL datapoint and function splits are illustrated in Figure B.1. We used it as follows:

- Meta-learning models: we meta-trained on the $ftrain, dtrain$ subset, sampled context points from the $ftest, dtrain$ subset, and reported metrics on all points from the $ftest, dtest$ subset.
- Transfer-learning models: we pre-trained on the $ftrain, dtrain$ subset and sampled points for fine-tuning from the $ftest, dtrain$ subset. We reported metrics on all points from $ftest, dtest$.
- Single-task models: we sampled points from $ftest, dtrain$ for training and reported metrics from $ftest, dtest$.

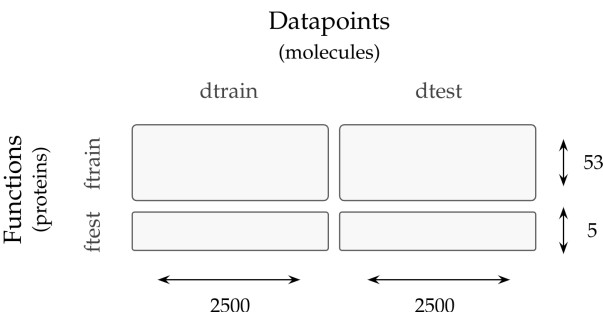

Figure B.1: DOCKSTRING subset

In the BO experiment we created an augmented dataset where scores were transformed either linearly (scalar multiplication and linear combination) or non-linearly (by taking the minimum or the maximum between a given score and the median of the score distribution for the corresponding protein target), and were combined with the quantitative estimate of drug-likeness (QED). We created 869 such functions; this odd number arose from creating an augmented dataset of 1000 functions and discarding those which included F2, JAK2 or LCK, since those three proteins were part of the objective functions (Appendix G).

Regarding the datapoint split, in our BO experiments we selected molecules from a library of 60k compounds that was created by sampling 30k molecules from the DOCKSTRING training set and the same amount from the DOCKSTRING test set. We meta-trained our models on the 869 augmented functions. We trained two MG-CNPs on these functions: one on the 2.5k training molecules from the FSL split, and another on the whole 60k molecules in the BO library.

### B.2 TARGET CORRELATIONS

We examined the maximum correlation of the 58 proteins in DOCKSTRING to any other protein in order to ascertain how challenging it would be to meta-generalize to them. We identified three

proteins from the test set that represented a range of dissimilarity to the meta-train set: PARP1 (highly correlated, very similar), ESR2 (medium correlation and similarity) and PGR (low correlation and similarity). We focused on these proteins to illustrate meta-generalization across a range of correlations.

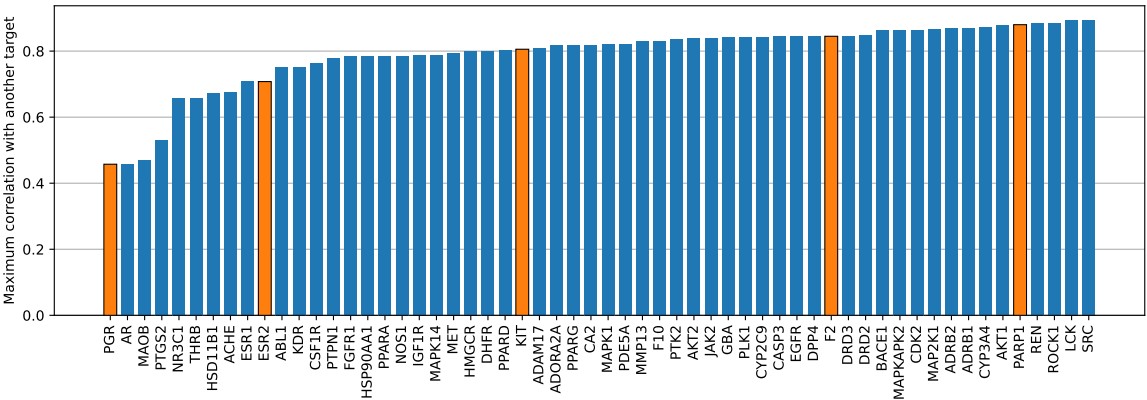

Figure B.2: Maximum Pearson correlation of each protein to any other protein in DOCKSTRING. Correlation is computed from docking scores in the full dataset of 260k molecules. Orange bars refer to test functions and blue ones to train functions.

### B.3 NUMBER OF OBSERVATIONS PER FUNCTION IN BIOACTIVITY DATASETS

FSL experiments in other domains (e.g. image classification) often focus on an extremely low number of context points, rarely benchmarking more than 100 contexts. In the context of drug discovery, however, the amount of observations per function in bioactivity datasets can vary widely, with many functions of interest having more than 100 labels. For example, in Figure B.3 we show the distribution of observations registered in protein binding affinity assays in ChEMBL33. Whether meta-learning can provide benefits in this data regime is an open research question. For this reason, we decided to investigate a wide range of context points, from 20 to 1000 (Table 1 and Appendix E).

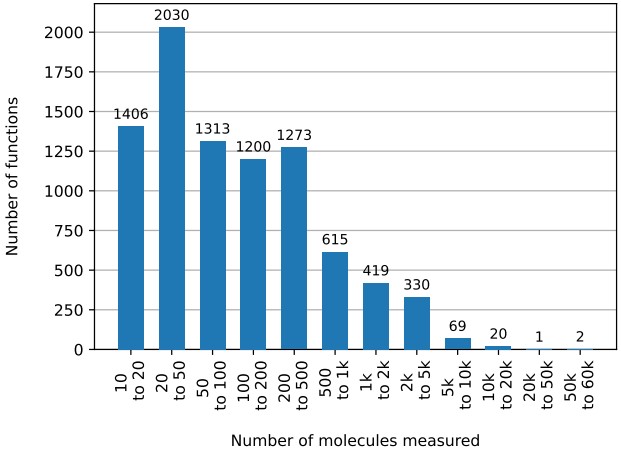

Figure B.3: Distribution of observations per function in the protein binding subset of ChEMBL33

## C   REPRODUCIBILITY

### C.1   REPEATABILITY

In every experiment we trained every model 10 times with different random initializations. We report the mean and standard error of these repetitions. The number of contexts in FSL experiments, as well as the initial points in the BO experiment, were also sampled randomly in each repetition.

### C.2   TRAINING DATA

The subset of points used for training each type of model (single task, transfer learning and meta-learning) is detailed in Appendix B.

### C.3   MODEL IMPLEMENTATION

#### C.3.1   MOLECULAR REPRESENTATIONS

We computed binary Morgan fingerprints (FPs) (Morgan, 1965) of length 1024 and radius 3 using RDKit (RDKit).

The connectivity of molecular graphs (MGs) was recreated from SMILES strings (Weininger, 1988) and atom and bond features were extracted using (RDKit). Atom features consisted of one-hot vectors indicating the atom type, the number neighbouring hydrogens, the number of neighbouring heavy atoms, the formal charge, the hybridisation type, whether the atom is placed within a ring, whether the atom is in an aromatic region and whether the atom is chiral. Atom features also included, as real numbers, the atomic mass, the Van der Waals radius and the covalent radius. Bond features included one-hot vectors with the bond type (single, double, tripe or aromatic), whether the bond is conjugated, whether it is part of a ring, and the stereoisometry type, if any.

#### C.3.2   MOLECULAR GRAPH ATTENTIVE ENCODER (MGAE)

All our models using MG representations rely on the molecular graph attentive encoder (MGAE). The architecture of MGAE is inspired by that of attentive FP (Xiong et al., 2020), but we perform pre-processing of the atomic and bond features with a small fully-connected neural network (FNN), post-processing of the final molecular representation with another small FNN, and speed up computation by changing the attention mechanism to query-key-value (QKV) (Vaswani et al., 2017) and by removing the GRU units (Chung et al., 2014). In particular, atoms and bond features are processed with fully-connected network of 2 layers and 50 hidden neurons in order to encode atomic and bond features into vectors of length 25. The encoding of all bonds connected to an atom are added and concatenated to the atomic encoding, which now has 50 elements. Then, we update the atomic encodings with 3 iterations of message passing from direct atomic neighbours, calculating attention coefficients with a QKV mechanism. The queries and keys are computed with a single-layer network. After the initial 3 iterations of message passing, a second stage begins where an imaginary "superatom" connected to all atoms is added to the graph, and another 3 iterations of message-passing are performed. The molecular representation is that of the superatom, which is passed through a small FNN to produce the final molecular representation, also of length 50.

#### C.3.3   NEURAL MODELS

All neural models were implemented in Pytorch (Paszke et al., 2019) and trained with the Adam optimizer using a learning rate of $10^{-3}$, weight decay of $10^{-5}$ and a cosine annealing scheduler.

#### C.3.4   SINGLE-TASK MODELS

*Random forest (RF).* Random forest regressors were implemented in scikit-learn (Buitinck et al., 2013) using the default parameters.

*Gaussian process (GP).* We implemented exact GPs in GPytorch (Gardner et al., 2018), using binary fingerprints and our own implementation of a Tanimoto kernel. The noise parameter was adjusted

automatically in GPytorch by evidence maximization for 2000 epochs. Automatic relevance determination (ARD) to find an optimal lengthscale for each feature was attempted but eventually rejected, since the large number of fingerprint features led to gross overfitting.

*Graph neural networks (GNNs).* Our GNN is a 6-layer FNN on top of the the MGAE, with maximum layer width of 1000 hidden neurons and layer normalization between each layer. We chose 6 layers to promote a fair comparison to our MG-NP models, which had an encoder and decoder of 3 layers each. GNNs were trained for 1000 epochs with a batch size of 50.

### C.3.5   TRANSFER-LEARNING MODELS

*GNN with random initialization (GNN random).* This model had the same architecture as the GNN in the previous section, but instead of training all layers, the GNN was initialized with random weights and only the last two layers were trained for 1000 epochs, while the rest were kept frozen, in a way reminiscent of a transfer learning experiment. In this way, the randomly-initalized layers worked as feature extractors.

*GNN pre-trained and fine-tuned (GNN fine-tuned).* This model had the same architecture as the GNN but was pre-trained as a multi-task model on the 53 meta-train proteins. Then, the last layer was changed (to make it single-task) and the last two layers were trained for 1000 epochs while the rest were kept frozen.

### C.3.6   MODEL-AGNOSTIC META-LEARNING (MAML)

In all experiments involving MAML (whether single-task GNN or NPs), we used a single step of gradient descent in the inner update with a learning rate of $10^{-3}$. We attempted other rates to try to increase training stability and decrease MAML's error bars but found this to be the most stable. In addition, our MAML training regime borrowed tricks from Antoniou et al. (2019) to improve robustness. In particular, we implement derivative-order annealing such that the first 50 epochs of training do first-order MAML and the rest do second-order MAML. We also implement cosine annealing of the outer learning rate. The tricks of multi-step loss optimization and per-step batch normalization are not applicable to our model since we performed a single gradient descent iteration in the inner loop. Due to memory constraints, we follow Raghu et al. (2020) and only apply MAML the last two layers of the encoder or the decoder of NPs, as explained in Section 5. Similarly, we only train the last two layers of single-task GNNs meta-trained with MAML.

### C.3.7   NEURAL PROCESSES

*Conditional neural processes (CNPs).* The FP-CNP had an encoder and decoder FNN of 3 layers each, with a maximum layer width of 1000, layer normalization between every layer, and an encoding size of 250. Layer normalization was chosen instead of batch normalization because, due to memory constraints, our batch size was just 2 functions. We choose the mean as commutative operation. The MG-CNP had the same architecture but built its encoder on top of the MGAE.

*Latent neural processes (LNPs).* The FP-LNP had the same encoder and decoder as the CNP, and an additional network to parameterize the mean and variance of the latent variable. This network was also a FNN of 3 layers with layer normalization. The size of the latent variable was 250. Again, we chose the mean as the commutative operation for aggregation. The MG-LNP had the same architecture but it built its encoder on top of the MGAE.

*Meta-training.* We trained using between 20 and 150 contexts and targets at each iteration. The amount of contexts and targets was itself sampled uniformly between 20 and 150. The contexts and targets were always disjoint, with the only exception of the experiment in Section 6.3 where they were allowed to overlap in order to be able to increase the size of both contexts and targets to 100% of the training set, thus increasing the number of effective epochs. In the FSL experiments, we trained for 3000 epochs on the 53 train proteins with a batch size of 2 functions per batch. In the BO experiments we trained for 1000 epochs on the augmented train set of combinations of docking scores with a batch size of 8.

*Adaptation during meta-testing.* The adaptation procedures with MAML and fine-tuning are described in Section 5.

*Loss functions.* As explained in Sections 3.1 and A.2, the parameters of the CNP $\psi = \{\theta\}$ were trained by backpropagation from the predictive log-likelihood

$$\mathcal{L}_\psi(y_t \mid x_c, y_c \,; x_t) = \log q_\theta(y_t \mid x_c, y_c \,; x_t).$$

The parameters of the LNP $\psi = \{\theta, \phi\}$ were trained by backpropagation from a loss function consisting of the predictive log-likelihood and a KL regularization term,

$$\mathcal{L}_\psi(y_t \mid x_c, y_c \,; x_t) = \log q_\theta(y_t \mid x_c, y_c \,; x_t) + D_{\mathrm{KL}}\big(q_\phi(z \mid x_d, y_d) \,\|\, q_\phi(z \mid x_c, y_c)\big).$$

Due to the extra regularization term, achieving convergence with LNPs is more challenging than with CNPs. In order to facilitate convergence, we tried training LNPs either with an exact KL divergence term or with an approximate one, and retained the best-performing model out of the two. We approximated the KL term

$$D_{\mathrm{KL}}\big(q_\phi(z \mid x_d, y_d) \,\|\, q_\phi(z \mid x_c, y_c)\big) = \mathbb{E}_{q_\phi(z \mid x_d, y_d)} \left[ \log \frac{q_\phi(z \mid x_d, y_d)}{q_\phi(z \mid x_c, y_c)} \right]$$

with Monte Carlo, taking a single sample from $q_\phi(z \mid x_d, y_d)$. We found that the FP-LNP achieved better performance when trained with an approximate KL regularization term, whereas the MG-LNP achieved better performance when trained with an exact one.

## D    THE CHALLENGE OF META-GENERALIZATION IN SINUSOIDS

In this experiment, we investigated the ability of meta-learning techniques to meta-generalize to 1D sinusoids that were slightly different from the ones seen during meta-training. We trained two models, a 6-layer fully-connected NN with MAML, and a fully-connected CNP with 3 encoder layers and 3 decoder layers (for implementation details, see Appendix C). Sinusoids had functional form

$$y = A \sin \big( f \, (x - B) \big),$$

where $A$ is the amplitude, $f$ the frequency and $B$ the shift. As meta-training set, we used $10^4$ functions with $A \in [0.1, 5)$, $f = 1.0$ and $B \in [0, \pi)$. All parameters were sampled uniformly from these ranges. For each function, we sampled points uniformly in $x \in [-5, 5]$ as contexts and targets. The context and target sets' sizes ranged between 5 and 25 and was also sampled too. The two models were trained for 10 epochs on the meta-train set. For meta-testing we used two different sets: in the first one, we created $10^4$ functions with the same parameters as the meta-train functions, and in the second we created $10^4$ functions with frequency $f = 1.5$. We sampled 20 points from each meta-test function in the same $x$ interval as contexts, and 1000 points as targets. We also evaluated the prediction error on the meta-train set in the same way. The models performed well on the meta-train set, as expected, and meta-generalized well to the test functions. However, they failed to meta-generalize to the test functions with the slightly different frequency D.1.

Table D.1: Mean squared error (MSE) of two meta-learning models on sinusoids. Values show mean and standard deviation.

|      | Train functions ($f = 1.0$) | Test functions ($f = 1.0$) | Test functions ($f = 1.5$) |
| ---- | --------------------------- | -------------------------- | -------------------------- |
| MAML | 0.39 (0.10)                 | 0.394 (0.10)               | 3.53 (0.28)                |
| CNP  | 0.01 (0.00)                 | 0.005 (0.00)               | 6.69 (0.17)                |

# E ADDITIONAL FEW-SHOT LEARNING (FSL) RESULTS

Table E.1: FSL of PARP1, a protein with high similarity to the meta-training set.

| | | PARP1 (high correlation), $R^2 \cdot 100$ | | | | | |
|---|---|---|---|---|---|---|---|
| | Context set size | 20 | 50 | 100 | 200 | 500 | 1000 |
| Single task | Dummy regressor | -13.6 (3.9) | -18.9 (3.0) | -19.6 (2.9) | -21.4 (1.7) | -21.9 (1.1) | -21.4 (0.6) |
| | FP-RF | 0.4 (2.1) | -1.2 (2.3) | 3.0 (2.5) | 15.1 (1.9) | 23.7 (0.8) | 30.9 (0.5) |
| | FP-GP | 6.4 (0.7) | 9.5 (1.7) | 15.6 (2.1) | 29.6 (1.3) | 47.5 (0.7) | 58.1 (0.3) |
| | GNN | **22.9 (11.9)** | **21.0 (13.9)** | **63.9 (4.2)** | **61.7 (9.0)** | **73.2 (0.6)** | **75.4 (0.5)** |
| Transfer learning | GNN (random) | 63.8 (1.4) | 69.9 (1.3) | 70.7 (2.5) | 73.7 (1.3) | **76.2 (0.7)** | **77.6 (1.3)** |
| | GNN (fine-tuned) | **75.0 (0.9)** | **77.3 (0.6)** | **77.9 (0.3)** | **76.2 (0.4)** | 75.4 (0.3) | 77.1 (0.2) |
| Meta-learning | GNN (MAML) | 51.6 (8.9) | 52.2 (9.0) | 52.0 (9.0) | 51.0 (9.8) | 50.9 (9.8) | 51.3 (9.8) |
| | ADKF-IFT | 10.8 (3.4) | 28.2 (1.8) | 38.8 (1.5) | 47.5 (1.1) | 55.2 (0.6) | 61.9 (0.3) |
| | FP-CNP | 55.5 (0.7) | 55.5 (0.5) | 55.4 (0.5) | 55.2 (0.4) | 55.4 (0.3) | 55.3 (0.4) |
| | FP-LNP | 35.1 (0.5) | 35.1 (0.5) | 35.1 (0.5) | 35.1 (0.5) | 35.1 (0.5) | 35.1 (0.5) |
| | MG-CNP | **81.6 (0.6)** | **82.7 (0.3)** | **83.0 (0.1)** | **83.1 (0.1)** | **83.1 (0.1)** | **83.0 (0.2)** |
| | MG-LNP | 81.4 (0.3) | 81.5 (0.3) | 81.6 (0.3) | 81.7 (0.3) | 82.0 (0.3) | 82.3 (0.3) |
| NPs with parameter adaptation | FP-CNP (MAML) | 57.8 (0.5) | 58.0 (0.6) | 58.3 (0.4) | 58.2 (0.3) | 58.2 (0.3) | 58.3 (0.3) |
| | FP-CNP (fine-tuned) | 55.3 (0.8) | 55.2 (0.7) | 55.7 (0.5) | 55.7 (0.4) | 56.1 (0.3) | 56.2 (0.2) |
| | FP-LNP (MAML) | 42.1 (0.6) | 42.3 (0.8) | 42.4 (0.7) | 42.4 (0.7) | 42.5 (0.8) | 42.4 (0.8) |
| | FP-LNP (fine-tuned) | 38.8 (1.3) | 36.0 (1.4) | 37.6 (1.4) | 40.3 (1.2) | 41.6 (1.2) | 41.2 (0.7) |
| | MG-CNP (MAML) | 82.1 (0.5) | 83.3 (0.3) | 83.5 (0.2) | 83.7 (0.2) | 83.7 (0.2) | 83.7 (0.2) |
| | MG-CNP (fine-tuned) | 82.1 (0.5) | **83.4 (0.2)** | **83.7 (0.2)** | **83.9 (0.2)** | **84.4 (0.2)** | **84.9 (0.1)** |
| | MG-LNP (MAML) | 80.8 (0.6) | 80.7 (0.6) | 80.8 (0.6) | 80.9 (0.6) | 81.2 (0.5) | 81.5 (0.5) |
| | MG-LNP (fine-tuned) | **82.5 (0.4)** | 82.3 (0.5) | 83.0 (0.3) | 83.5 (0.3) | 84.2 (0.2) | 84.7 (0.2) |

| | | PARP1 (high correlation), NLPD | | | | | |
|---|---|---|---|---|---|---|---|
| | Context set size | 20 | 50 | 100 | 200 | 500 | 1000 |
| Single task | FP-GP | **0.85 (0.01)** | **0.71 (0.01)** | **0.64 (0.01)** | **0.55 (0.01)** | **0.40 (0.01)** | **0.28 (0.00)** |
| Meta-learning | ADKF-IFT | **0.73 (0.04)** | **0.57 (0.02)** | **0.48 (0.02)** | **0.39 (0.01)** | **0.31 (0.01)** | **0.22 (0.00)** |
| | FP-CNP | 3.61 (0.11) | 3.50 (0.11) | 3.52 (0.07) | 3.49 (0.06) | 3.49 (0.05) | 3.50 (0.06) |
| | FP-LNP | 1.46 (0.01) | 1.46 (0.01) | 1.46 (0.01) | 1.46 (0.01) | 1.46 (0.01) | 1.46 (0.01) |
| | MG-CNP | 0.83 (0.06) | 0.78 (0.02) | 0.75 (0.01) | 0.74 (0.01) | 0.74 (0.01) | 0.74 (0.01) |
| | MG-LNP | 0.75 (0.01) | 0.75 (0.01) | 0.75 (0.01) | 0.75 (0.01) | 0.74 (0.01) | 0.74 (0.01) |
| NPs with parameter adaptation | FP-CNP (MAML) | 3.42 (0.17) | 3.35 (0.12) | 3.31 (0.08) | 3.27 (0.06) | 3.18 (0.05) | 3.19 (0.03) |
| | FP-CNP (fine-tuned) | 3.76 (0.84) | 3.20 (0.25) | 3.46 (0.27) | 3.49 (0.22) | 3.46 (0.10) | 3.48 (0.11) |
| | FP-LNP (MAML) | 1.44 (0.02) | 1.43 (0.02) | 1.43 (0.02) | 1.43 (0.02) | 1.43 (0.02) | 1.43 (0.02) |
| | FP-LNP (fine-tuned) | 2.14 (0.08) | 1.87 (0.02) | 2.68 (0.05) | 3.24 (0.09) | 3.52 (0.09) | 3.66 (0.11) |
| | MG-CNP (MAML) | 0.80 (0.03) | **0.75 (0.02)** | **0.73 (0.01)** | **0.72 (0.00)** | **0.71 (0.00)** | 0.71 (0.00) |
| | MG-CNP (fine-tuned) | 0.95 (0.08) | 0.84 (0.05) | 0.79 (0.03) | 0.76 (0.01) | 0.73 (0.01) | 0.70 (0.01) |
| | MG-LNP (MAML) | **0.78 (0.02)** | 0.78 (0.02) | 0.78 (0.02) | 0.78 (0.02) | 0.77 (0.02) | 0.76 (0.02) |
| | MG-LNP (fine-tuned) | 0.91 (0.07) | 0.81 (0.03) | 0.77 (0.02) | 0.74 (0.01) | 0.72 (0.01) | **0.69 (0.01)** |

Table E.2: FSL of PGR, a protein with low similarity to the meta-training set.

| | | PGR (low correlation), $R^2 \cdot 100$ | | | | | |
|---|---|---|---|---|---|---|---|
| Single task | Dummy regressor | -13.3 (4.6) | -5.3 (1.9) | -2.7 (0.8) | -1.6 (0.4) | -1.6 (0.3) | -1.5 (0.2) |
| | FP-RF | -13.2 (8.7) | -4.7 (2.2) | -0.3 (1.4) | 2.0 (1.3) | **8.9 (0.8)** | 11.9 (0.5) |
| | FP-GP | **-11.3 (4.9)** | **1.0 (1.8)** | **6.2 (0.9)** | **8.4 (0.7)** | **8.9 (0.6)** | 11.2 (0.8) |
| | GNN | -56.6 (33.0) | -13.0 (6.6) | -4.4 (2.6) | -1.9 (4.8) | 6.7 (1.6) | **17.4 (3.0)** |
| Transfer learning | GNN (random) | -21.2 (16.9) | 14.5 (2.6) | 9.9 (3.8) | 13.5 (3.9) | **23.3 (1.6)** | **29.9 (0.7)** |
| | GNN (fine-tuned) | **21.1 (1.8)** | **26.4 (1.2)** | **23.7 (1.4)** | **16.3 (1.1)** | 19.4 (1.0) | 24.7 (1.0) |
| Meta-learning | GNN (MAML) | -31.2 (7.7) | -26.6 (6.2) | -28.4 (6.3) | -30.5 (6.4) | -32.4 (6.8) | -33.0 (6.4) |
| | ADKF-IFT | -6.6 (4.1) | 1.7 (1.4) | 4.5 (0.7) | 5.2 (0.8) | 8.7 (0.9) | 14.6 (0.4) |
| | FP-CNP | -36.7 (4.7) | -37.5 (4.5) | -39.1 (3.8) | -39.7 (2.3) | -42.4 (2.9) | -41.6 (2.5) |
| | FP-LNP | -70.4 (1.8) | -70.4 (1.8) | -70.4 (1.8) | -70.4 (1.8) | -70.4 (1.8) | -70.4 (1.8) |
| | MG-CNP | **19.1 (2.5)** | **23.2 (1.7)** | **25.6 (1.4)** | **27.5 (1.5)** | **27.5 (1.2)** | **27.2 (1.3)** |
| | MG-LNP | 2.4 (3.7) | 3.0 (3.7) | 3.8 (3.7) | 5.5 (3.8) | 10.6 (3.7) | 15.7 (3.4) |
| NPs with parameter adaptation | FP-CNP (MAML) | -26.4 (6.0) | -19.0 (2.3) | -19.0 (1.8) | -20.7 (1.6) | -20.2 (1.6) | -19.9 (1.6) |
| | FP-CNP (fine-tuned) | -24.5 (5.2) | -13.7 (1.5) | -9.2 (1.3) | -5.7 (1.5) | 1.6 (1.1) | 7.7 (0.8) |
| | FP-LNP (MAML) | -24.1 (7.9) | -21.2 (6.7) | -21.0 (6.4) | -21.6 (6.5) | -21.3 (6.3) | -20.9 (6.3) |
| | FP-LNP (fine-tuned) | -1.2 (3.4) | 3.6 (0.6) | 7.7 (0.8) | 8.3 (0.6) | 7.3 (0.9) | 6.2 (0.6) |
| | MG-CNP (MAML) | **22.2 (2.2)** | 29.7 (1.5) | 31.3 (1.6) | 32.6 (1.4) | 33.3 (1.1) | 33.4 (1.2) |
| | MG-CNP (fine-tuned) | 20.3 (3.9) | **31.9 (1.4)** | **36.6 (1.2)** | **40.7 (0.8)** | **43.7 (0.2)** | **45.1 (0.3)** |
| | MG-LNP (MAML) | 10.2 (3.1) | 11.5 (2.8) | 12.2 (2.7) | 13.2 (2.7) | 16.0 (2.4) | 19.7 (1.9) |
| | MG-LNP (fine-tuned) | 10.8 (9.3) | 7.7 (6.5) | 18.3 (3.7) | 31.6 (2.0) | 40.1 (1.5) | 42.2 (1.6) |

| | | PGR (low correlation), NLPD | | | | | |
|---|---|---|---|---|---|---|---|
| Single task | FP-GP | **0.64 (0.04)** | **0.49 (0.02)** | **0.43 (0.01)** | **0.41 (0.01)** | **0.40 (0.00)** | **0.38 (0.00)** |
| Meta-learning | ADKF-IFT | **0.52 (0.03)** | **0.44 (0.01)** | **0.42 (0.01)** | **0.41 (0.01)** | **0.40 (0.01)** | **0.36 (0.00)** |
| | FP-CNP | 7.29 (1.16) | 8.03 (1.17) | 7.83 (0.79) | 7.44 (0.45) | 8.16 (0.58) | 8.03 (0.44) |
| | FP-LNP | 1.72 (0.01) | 1.72 (0.01) | 1.72 (0.01) | 1.72 (0.01) | 1.72 (0.01) | 1.72 (0.01) |
| | MG-CNP | 1.18 (0.02) | 1.13 (0.02) | 1.09 (0.01) | 1.07 (0.01) | 1.07 (0.01) | 1.08 (0.01) |
| | MG-LNP | 1.28 (0.03) | 1.27 (0.03) | 1.27 (0.03) | 1.25 (0.03) | 1.21 (0.03) | 1.16 (0.03) |
| NPs with parameter adaptation | FP-CNP (MAML) | 4.60 (0.52) | 5.18 (0.49) | 5.13 (0.40) | 5.09 (0.29) | 5.23 (0.26) | 5.31 (0.23) |
| | FP-CNP (fine-tuned) | 2.36 (0.28) | 2.23 (0.15) | 1.69 (0.04) | 1.49 (0.01) | 1.45 (0.01) | 1.44 (0.01) |
| | FP-LNP (MAML) | 1.49 (0.04) | 1.46 (0.03) | 1.46 (0.03) | 1.46 (0.03) | 1.46 (0.03) | 1.46 (0.03) |
| | FP-LNP (fine-tuned) | 1.37 (0.02) | 1.34 (0.01) | 1.32 (0.01) | 1.32 (0.00) | 1.46 (0.02) | 1.62 (0.03) |
| | MG-CNP (MAML) | **1.14 (0.02)** | **1.06 (0.01)** | 1.04 (0.01) | 1.03 (0.01) | 1.02 (0.01) | 1.02 (0.01) |
| | MG-CNP (fine-tuned) | 1.15 (0.02) | **1.06 (0.01)** | **1.03 (0.01)** | **1.00 (0.01)** | **0.98 (0.01)** | **0.96 (0.01)** |
| | MG-LNP (MAML) | 1.20 (0.02) | 1.19 (0.02) | 1.19 (0.02) | 1.18 (0.02) | 1.16 (0.02) | 1.14 (0.02) |
| | MG-LNP (fine-tuned) | 1.30 (0.11) | 1.29 (0.07) | 1.22 (0.05) | 1.06 (0.01) | 1.01 (0.01) | 0.98 (0.01) |

# F RANDOM SAMPLING OF OBSERVATIONS PROTECTS NPS FROM OVERFITTING

Increasing the percentage of points sampled as contexts or targets leads to less unique function views and to more effective epochs. This causes overfitting, both in terms of $R^2$ (Figure F.1) and NLPD (see Figure 4).

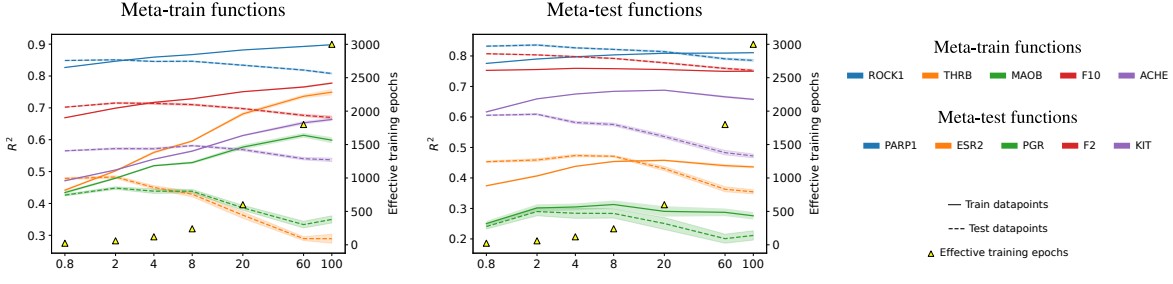

Figure F.1: Increasing the percentage of points sampled as contexts or targets at each iteration leads to memorization of the labels from the train points of the train functions (left, solid lines) and a degradation of performance on the test points of the train functions (left, dashed line) and all points of the test functions (right, solid and dashed lines).

As the percentage of points sampled as contexts or targets increased, the calibration of the uncertainty estimates deteriorated substantially (Figure F.2).

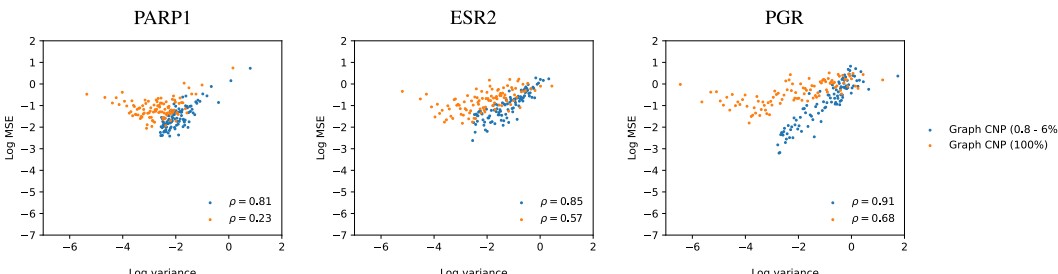

Figure F.2: A MG-CNP meta-trained using a small fraction of the training points as contexts or targets at each iteration (blue) maintains a good calibration, as shown by the high Pearson correlation between the predicted log variance and the prediction error. In contrast, a MG-CNP meta-trained using all training points as context or targets (orange) displays poor calibration.

## G  BAYESIAN OPTIMIZATION OBJECTIVES

Our BO experiments (Section 6.4) used the objective functions druglike F2 and selective JAK2 from the DOCKSTRING optimization benchmark (García-Ortegón et al., 2022).

- *Druglike F2* is a comparatively easy task that requires docking well to a single target and satisfying orally-bioavailable druglike properties according to QED:

$$f_{\text{F2}}(\ell) = s(\ell, \text{F2}) + 10\big(1 - \text{QED}(\ell)\big),$$

where $\ell$ is a ligand molecule, and $s(\ell, p)$ is the docking score of ligand $\ell$ against protein $p$.

- *Selective JAK2* is a comparatively difficult taks that requires docking well against JAK2 and not against LCK. This is a hard task since the docking scores of these two kinases are very highly correlated. This objective reflects a real interest in the drug discovery community to design selective kinase inhibitors.

$$f_{\text{JAK2}}(\ell) = s(\ell, \text{JAK2}) - \min\big(s(\ell, \text{LCK}) - 8.1,\, 0\big) + 10\big(1 - \text{QED}(\ell)\big)$$

(Note that $-8.1$ is the median of LCK docking scores.)

