In Figure 2 we examine the calibration of the MG-CNP and MG-LNP from Section 6.1, using the same data split as before (Appendix B). We ranked the 2.5k test datapoints by predicted variance from most confident (lowest predicted variance) to most uncertain (highest predicted variance), and partitioned them into 100 groups of 25 datapoints, which we call confidence percentiles. Lower percentiles represent higher confidence. We computed the average mean square error (MSE) and the average predicted variance within each percentile. Then, we inspected calibration with a single context set size (scatter plots) or across a range of context set sizes (heatmaps).

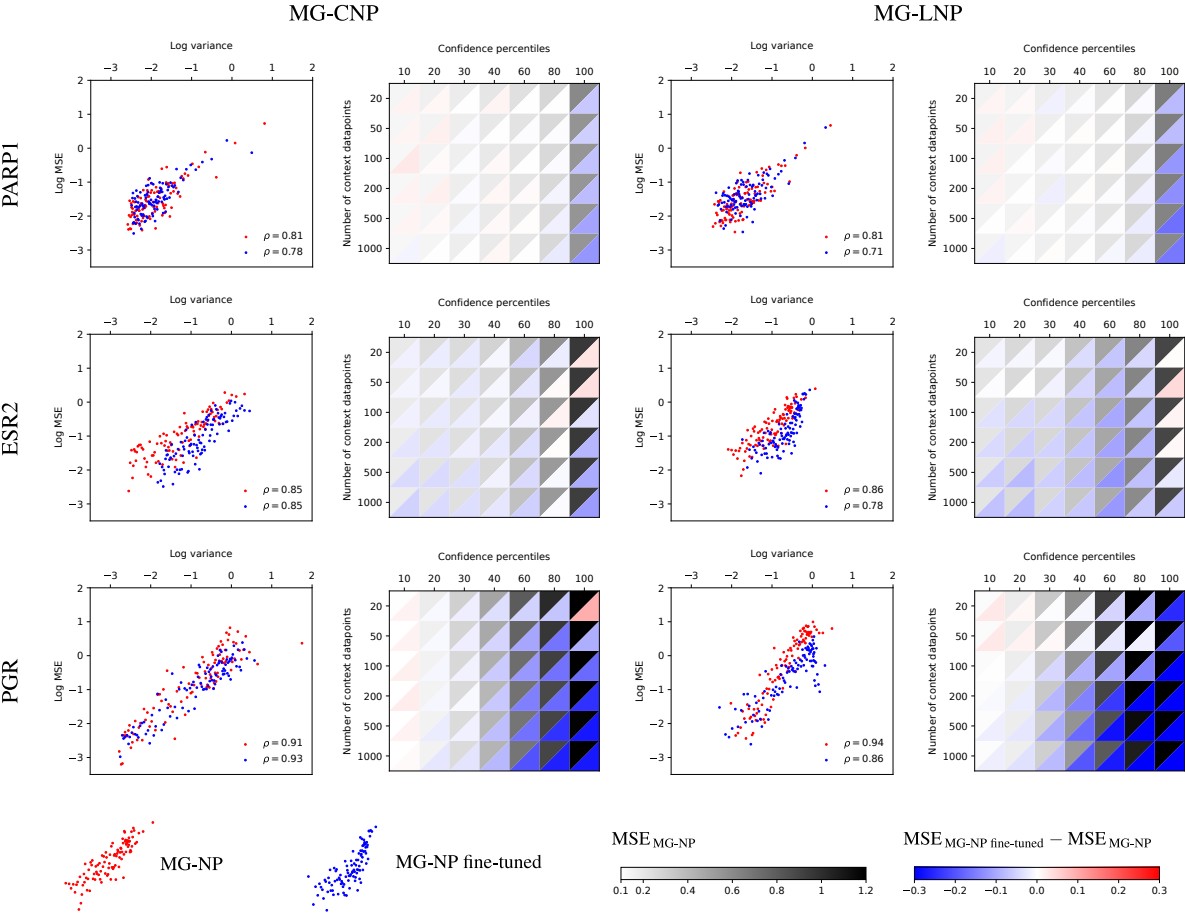

Figure 2: Calibration of uncertainty estimates from MG-CNP and MG-LNP. Scatterplots on the left show the log mean MSE and log mean predicted variance in each confidence percentile (percentiles of target datapoints ranked by predicted variance). The right heatmaps depict the absolute MSE of MG-NPs (black and white scale; black means higher error) and the MSE difference between the fine-tuned and the unmodified models (blue and red scale; blue indicates that fine-tuned is better).

The scatter plots on the left columns of Figure 2 show the log average MSE and predicted variance of the confidence percentiles of two models: an unmodified NP (i.e. without parameter adaptation) in red and a fine-tuned NP in blue. Both models use 200 datapoints from the training set as contexts. The log MSE and log variance correlate highly, with Pearson correlation $\rho > 0.7$, suggesting good calibration.

The heatmaps on the right columns of Figure 2 show the MSE of unmodified and fine-tuned models in the meta-test tasks, using context sets ranging in size from 20 to 1000. The row dimension indicates context set size and the column dimension shows test datapoints grouped by confidence percentile (the first column includes the 1st to the 10th percentile, the second the 11th to the 20th, etc). In each cell, the black and white scale represents the MSE of the unmodified model. As before, we observe that MG-NPs are well calibrated, with higher errors in the higher percentiles[2]. The red and blue scale indicates the difference between the MSE of the fine-tuned and the unmodified models. Blue cells signify that the fine-tuned model beat the unmodified one in that context size and confidence pair. In general, most fine-tuning gains came from the most erroneous unmodified predictions. This is to be expected: regions of the input space where predictions have higher loss will provide a larger signal for parameter adaptation.

### 6.3 RANDOMIZATION OF CONTEXT AND TARGET SETS PROTECTS NPS FROM OVERFITTING

The MG-CNP, a model with millions of parameters, exhibited good uncertainty calibration in meta-test functions despite having been trained with a maximum likelihood objective in as few as 2.5k molecules. Even though the CNP objective lacks a regularization term like that of the LNP, we did not find evidence that the model overfits, in the sense of underestimating posterior uncertainty. We hypothesized that this may be due to the randomization of context and target points during meta-training. The effect of this process may be two-fold: first, random sampling induces a combinatorially large number of unique views of each function, which may resemble an augmentation of the function set; and second, by using only a subset of all observations at each epoch, the number of effective epochs is kept low, hence reducing the risk of overfitting to any single datapoint.

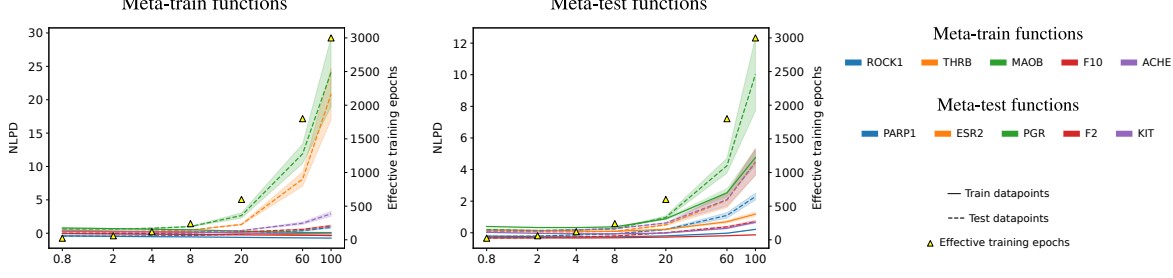

Figure 3: Random sampling of observations during meta-training protects from overfitting. Increasing the percentage of points sampled as contexts or targets leads to less unique function views and to more effective epochs. This causes overfitting, with memorization of the labels from the train points of the train functions (left, solid lines) and a degradation of performance on the test points of the train functions (left, dashed line) and all points of the test functions (right, solid and dashed lines).

In previous experiments, we trained MG-NPs using between 0.8% and 6% of observations as contexts and targets at each iteration. To investigate if random subsampling during meta-training protects NPs from overfitting, we trained a collection of MG-CNPs with the same architecture and on the same dataset as before, but using increasing fractions of observations as contexts and targets. As the fraction of points sampled growed, MG-CNPs overfit to the training molecules of the train functions, both in terms of NLPD (Figure 3) and $R^2$ (Appendix F). This suggests that, in order to achieve adequate meta-generalization and calibration, it is critical to tune the size of the context and target sets.

---

[2]Increasing the context set size (i.e. moving from top to bottom in the heatmaps of Figure 2) also improved performance and decreased MSE, as shown in Table 1 of the previous section and in Appendix E. However, the difference was small compared to the difference across confidence percentiles, so the latter dominates the black and white color scale.

## 6.4 BAYESIAN OPTIMIZATION (BO)

We evaluated the MG-CNP in sequential learning using two objective functions from DOCK-STRING