# OpenReview forum: "Graph neural processes and their application to molecular functions"
_ICLR.cc/2024/Conference — Submitted to ICLR 2024_

### Official Review · Reviewer_cRU3 · 2023-10-30

**Soundness:** 3 good
**Presentation:** 2 fair
**Contribution:** 2 fair
**Rating:** 3
**Confidence:** 4

**Summary:**

The paper proposes an approach based on graph neural processes for meta learning for drug discovery. The authors suggest replacing the MLP encoder of vanilla latent/deterministic neural processes with a graph neural network in order to capture higher-order interactions between the input covariables which in this case are atomic and atom-atom bond features. In addition, they propose a fine-tuning approach to adapt parameters after meta-training and an adoption of model-agnostic meta learning for NPs.

**Strengths:**

- Phrasing the problem of drug discovery as a meta-learning problem and using graph neural networks as encoders for neural processes is in the reviewer's opinion both original and reasonable.
- The paper is well written and easy to follow, and the problems of meta-learning in drug discovery is well delineated.
- The approach of fine-tuning is intuitive and seems reasonable.

**Weaknesses:**

- The main contribution of the paper is the usage of vanilla NPs with an encoder that is an adapted graph neural network of a previously introduced method [1] . In total, the contribution seems too incremental and too little.
- The description of the methodology itself (molecular graph attentive encoder) is not detailed enough and very superficial (in total 5 lines of the entire manuscript).
- The fine-tuning and MAML approaches for parameter adaption described in the paper are of little novelty. In addition, the theoretical benefit and motivation of the MAML tuning is not clear to the reviewer since NPs can generally already be considered as meta learnerns.? Empirically, the MAML tuning sometimes improves and sometimes worsens predictive performance (see, e.g., Table~1 FP-CNP with FP-CNP (MAML) or MG-LNP with MG-LNL (MAML)).
- The experimental section seems very thin and more evaluations with missing competing methods should be made. See, e.g., [2] as a reference.
- The reference section is incomplete and sometimes incorrect. For instance, the "Attention is all you need" paper is from 2017 and not from 2023 and misses the conference information.
- The authors fail to cite relevant literature on graph neural processes, e.g. [3].

[1] https://pubs.acs.org/doi/10.1021/acs.jmedchem.9b00959
[2] https://arxiv.org/abs/2205.02708
[3] https://arxiv.org/abs/2305.18719

**Questions:**

- Some clarifications of the math of the encoder structure or an illustrative figure would in the reviewer's opinion improve the quality of the manuscript. While background on NPs is explained in sufficient detail (both in the main manuscript as well as the appendix), the actual method is not described at all.
- The authors could evaluate the case where a NP has both a latent and deterministic encoder. See, e.g., [2]
- As far as I can tell, the authors do not compare themselves against recent methods such as in [1]. Is this true and if so is there any reason for that?
- LNPs are generally harder to train then CNPs. Is the poor performance of LNPs due to this fact or how can it be explained? Is it because the authors seemed to have trained only for a fixed number of iterations and not until converge (see Appendix C3.7)?

[1] https://arxiv.org/abs/2205.02708
[2] https://arxiv.org/abs/1901.05761

---

> ### Author Response · Authors · 2023-11-23
>
> We thank Reviewer cRU3 for their time and comments, to which we reply below.
>
> **Insufficient description of the GNN architecture**
>
> We thank the reviewer for raising this point. We would like to explain that the architecture of the GNN used by our MG-NPs is based on a very popular architecture (Attentive FP [1]) whose paper we reference and can be checked by the readers. The adaptation procedure is a bit trickier to replicate than the architecture (especially the one using MAML), so we used more space on that section instead, given the length constraints. We explain the differences between Attentive FP and our GNN in Appendix C.3.2. In addition, if our article is accepted we will publish our code with the camera-ready version.
>
> **Fine-tuning and MAML for parameter adaptation are of little novelty.**
>
> As far as we know, we are the first to propose parameter adaptation for NPs during meta-testing. The reason why parameter adaptation could be helpful is that the meta-learning done by a NP, and the adaptation based on the contexts, are not necessarily perfect. Unlike a Gaussian process, which will make accurate predictions on the context points it has seen during its evaluation, a NP has no theoretical guarantees of accuracy on the context points. We exploit context predictions' errors to improve adaptation to meta-test functions during meta-testing, thus helping to solve the issue of the applicability of NPs when the meta-train and meta-test functions diverge.
>
> **Incorrect citations in bibliography**
>
> We thank the reviewer for catching this error! We have investigating the source and it appears the mistaken citations had been downloaded from arXiv. For example, the button "Export BibTeX citation" on the arXiv page of "Attention is all you need" ( https://arxiv.org/abs/1706.03762 ) yields a citation without the conference information and with the year 2023. We were unaware arXiv worked in this way. We apologize for this error, which we have now amended.
>
> **Not citing paper "Graph Neural Processes for Spatio-Temporal Extrapolation" by Hu et al**
>
> We thank the reviewer for pointing out this paper, which we were not aware of. Even though the names of the two models are similar, this model is actually very different from ours, since it is designed to learn node signals within a graph, and ours is designed to learn graph signals across many graphs. Therefore, their model is not readily applicable to our task. In addition, their application (weather prediction) is very different from ours (molecular property prediction). Because of these reasons it does not seem appropriate to cite this paper. However, if after reading our explanation the reviewer believes otherwise, we would be happy to reconsider.
>
> **Why would MAML benefit NPs if both are meta-learning methods?**
>
> Indeed, it may be surprising to combine MAML and NPs since both are meta-learning methods. However, as described above, NPs may not predict the context points accurately, which provides an opportunity for parameter adaptation using the loss of the context points. Such adaptation could be done in multiple ways. One is fine-tuning for multiple epochs, and another one is taking a single iteration of gradient descent after having trained on MAML. Here, we simple use MAML as a strategy to achieve fast adaptation. The meta-training regime of a NP trained with MAML is a bit tricky: it involves computing the loss of the NP's context predictions, taking one iteration of gradient descent on the context loss (the MAML inner update), computing the loss of the NP's target predictions, and taking one iteration of gradient descent through the inner update (the MAML outer update).
>
> **Why does MAML sometimes improve and sometimes worsen performance?**
>
> MAML always improves the performance of a single-task model in the low-data regime (Table 1, GNN with MAML vs GNN with just 20 context points), as expected. However, MAML doesn't always improve the performance of NPs. This may be because of training instabilities, an issue that is well documented for MAML [2]. We tried to improve robustness by implementing improvements from MAML++ but we still found high variance across different random seed initializations. Therefore, using MAML on NPs brings about a potential trade-off: MAML may improve the performance of NPs by improving adaptation to meta-test functions by taking a single step of gradient descent, but MAML may also worsen performance by hindering learning during meta-training due to training instabilities.
>
> **References**
>
> [1] Xiong et al 2020. Pushing the boundaries of molecular representation for drug discovery with the graph attention mechanism https://pubs.acs.org/doi/10.1021/acs.jmedchem.9b00959
>
> [2] Antoniou et al 2018. How to train your MAML. https://arxiv.org/abs/1810.09502
>
> [3] Chen et al 2022. Meta-learning adaptive deep kernel gaussian processes for molecular property prediction. https://arxiv.org/abs/2205.02708

---

> ### Author Response · Authors · 2023-11-23
>
> **Comparison with ADKF-IFT**
>
> We thank the reviewer for raising this point. ADKF-IFT [3] is now included as a baseline.
>
> **Why do LNPs perform worse than CNPs?**
>
> Initially, we trained CNPs and LNPs for the same amount of iterations in order to promote an equalitarian allocation of resources and a fair comparison. However, we believed the LNP did not perform so well partly because its regularization term promoted underfitting and made reaching a good optimum challenging. Therefore, we attempted trainining during twice the number of iterations (6000 epochs). However, after 3000 epochs the results did not change substantially. In addition, in order to facilitate reaching a good optimum, we trained two versions of each LNP: one where the regularization term of the ELBO-like objective was computed analytically, and one where the regularization term was approximated with a single Monte Carlo sample (Appendix C.3.7). Then, we chose the version which performed best. At this point, we felt that further optimizing the LNP preferentially would have made the comparison unfair towards other models. All in all, we spent considerably more time and effort optimizing the LNP than the CNP.
>
> **References**
>
> [1] Xiong et al 2020. Pushing the boundaries of molecular representation for drug discovery with the graph attention mechanism https://pubs.acs.org/doi/10.1021/acs.jmedchem.9b00959
>
> [2] Antoniou et al 2018. How to train your MAML. https://arxiv.org/abs/1810.09502
>
> [3] Chen et al 2022. Meta-learning adaptive deep kernel gaussian processes for molecular property prediction. https://arxiv.org/abs/2205.02708

---

### Official Review · Reviewer_DH7x · 2023-10-31

**Soundness:** 3 good
**Presentation:** 4 excellent
**Contribution:** 3 good
**Rating:** 6
**Confidence:** 3

**Summary:**

Meta-learning is crucial in fields such as biology, where a variety of test functions exist and sparse data is typical. Additionally, uncertainty measures are typically of interest, to aid in deciding which of the predictions should undergo costly experimental validation. In this work, the authors benchmark deep neural process, a type of deep models that also model uncertainty, for few-shot learning. The authors show that even small modifications to the test functions can massively affect meta-generalization, and use two approaches to address this: fine-tuning and a single step of gradient descent on a MAML-trained neural process. They benchmark neural processes in DOCKSTRING, a dataset of docking scores of 260k ligands against 58 diverse proteins, using molecular fingerprints and graphs as input representations.

**Strengths:**

-	The work is nice and easy to follow
-	Elegant and simple experiment to show the disruption of meta-generalization with divergent test functions (Figure 1)
-	Provides useful take-aways in few-shot learning experiments

**Weaknesses:**

- Restricted evaluation to DOCKSTRING

**Questions:**

Surprisingly, I don’t have any questions regarding the work itself. It was very clear, easy to follow, and thorough in the evaluation of deep NPs for few-shot learning in DOCKSTRING. I believe this is an important work in benchmarking deep NPs that would be of great use to the community.

---

> ### Author Response · Authors · 2023-11-23
>
> We thank Reviewer DH7x for their time and comments. We have now added ADKF-IFT as an additional baseline, and have added one figure (Figure 2) where we compare its calibration with that of NPs and a GP on fingerprints.

---

### Official Review · Reviewer_PKuZ · 2023-11-01

**Soundness:** 2 fair
**Presentation:** 2 fair
**Contribution:** 2 fair
**Rating:** 5
**Confidence:** 4

**Summary:**

This paper studies meta-learning approaches for molecular tasks, and focuses on introducing neural process (NP) for this application. Apart from building different NP models (CNP,LNP) for molecules, taking fingerprints (FPs) or molecular graphs (MGs) as input features, this paper emphasizes the challenge of meta-generalization in molecular tasks. To close to real world molecular applications, it sets up experiments with an unusual meta-learning setting: the correlation between training and testing tasks are controlled at a low degree, and the size of context varies in a large range. To deal with, this paper proposes to combine gradient-based adaptation (MAML, fine-tuning) with NP model. The authors tailor DOCKSTRING dataset, and detail empirical results show that MG-CNPc(fine-tuned) has a performance advantage in most cases.

**Strengths:**

1.	This paper comprehensively study NP-based models on molecular tasks, including different NP variants, different molecular features, different additional adaptation strategies.

2.	It pointed out the challenge that tasks are highly diverse in real world molecular applications. And propose additional adaptation steps should be adopted based on NP models to increase the meta-generalizability.

3.	Data processing and empirical results are shown in detail. It looks convincing that the proposed method could show advantage with such setting.

**Weaknesses:**

1. Lack of novelty. As a representative amortized meta-learner, NP has been widely studied. This paper adopts the most conventional NP models on molecular tasks. “NP+gradient steps” is also a popular way to improve meta-learning performance by combining two adaptation strategies. (in similar fields, there is [1]). It seems little technical contribution in this paper.

2. The authors propose that existing datasets are highly homogeneous across tasks, while in reality the task diversity should be considered. However, there lacks evidence in this paper. No empirical results of existing popular datasets (e.g., fs-mol[2], moleculenet[3]), nor comparing them with real-world cases are provided.

3. Lack of benchmark datasets and baselines. Since the proposed is following a standard meta-learning setting, existing few-shot molecular property prediction methods [4,5,6], should be considered. Among them, [5] is also applicable for regression task, which should be compared on DOCKSTRING. And the proposed method should also be applicable for classification tasks, so it should be tested on [2][3], and compared with [4,5,6].

4. Poor organization of related works. The related works mix everything (i.e. datasets, methods) together, which are hard to read.

[1] Zhang, Q., Zhang, S., Feng, Y., & Shi, J. (2023). Few-Shot Drug Synergy Prediction With a Prior-Guided Hypernetwork Architecture. IEEE Transactions on Pattern Analysis and Machine Intelligence.

[2] Stanley, M., Bronskill, J. F., Maziarz, K., Misztela, H., Lanini, J., Segler, M., ... & Brockschmidt, M. (2021, August). Fs-mol: A few-shot learning dataset of molecules. In Thirty-fifth Conference on Neural Information Processing Systems Datasets and Benchmarks Track (Round 2)

[3] Wu, Z., Ramsundar, B., Feinberg, E. N., Gomes, J., Geniesse, C., Pappu, A. S., ... & Pande, V. (2018). MoleculeNet: a benchmark for molecular machine learning. Chemical science, 9(2), 513-530.

[4] Wang, Y., Abuduweili, A., Yao, Q., & Dou, D. (2021). Property-aware relation networks for few-shot molecular property prediction. Advances in Neural Information Processing Systems, 34, 17441-17454.

[5] Chen, W., Tripp, A., & Hernández-Lobato, J. M. (2022, September). Meta-learning adaptive deep kernel gaussian processes for molecular property prediction. In The Eleventh International Conference on Learning Representations.

[6] Schimunek, J., Seidl, P., Friedrich, L., Kuhn, D., Rippmann, F., Hochreiter, S., & Klambauer, G. (2023). Context-enriched molecule representations improve few-shot drug discovery. arXiv preprint arXiv:2305.09481.

**Questions:**

Please refer to weaknesses.

---

> ### Author Response · Authors · 2023-11-23
>
> We would like to thank reviewer PKuZ for their time. We reply to their comments below.
>
> **Showing lack of heterogeneity in current benchmarks**
>
> We would like to explain that when we referred to lack of heterogeneity in current benchmarks, we were referring to the toy datasets used by the broader meta-learning community, not to FS-Mol and MoleculeNet. Examples of the types of datasets we were referring to are the 1D sine function used in the original MAML paper [1], MNIST in the original CNP and LNP publications [2,3], or samples from a Gaussian process model with slightly different hyperparameters (the latter is often used in the NP literature, see e.g.[4] for a recent example). These simple datasets may be useful to develop novel meta-learning models but they do not represent the degree of heterogeneity of scientific datasets. Therefore, models that do well on such datasets may fail catastrophically when faced with divergent meta-train and meta-test functions. We showed this was the case for MAML and the CNP in Figure 1, where even a small change in the 1D sine function led to loss of meta-generalization.
>
> **Comparison with other real-world datasets**
>
> We thank the reviewer for raising this point, which is a fair one. We made a decision to focus on benchmarking docking scores in this publication and leave bioactivity endpoints for a future, more-biologically-focused publication. There are three compelling reasons to use docking scores when benchmarking drug-discovery algorithms. First, some widely-used bioactivity datasets such as MoleculeNet may present their own flaws, e.g. they present intra-function inconsistencies due to pooling from different publications [5,6]. Second, docking scores are highly useful for drug discovery, which makes them interesting to predict with ML by their own right. Some recent high-impact publications that had ML-prediction of docking scores at their core are [7,8] . Finally, docking depends on the simulation of a 3D interaction between a protein and a molecular ligand. Therefore, a model that is able to learn docking scores must be sophisticated enough to capture the structural relation between the protein target and the molecule. In fact, some docking scores are very challenging to predict, e.g. in the original publication of the dataset we are using (DOCKSTRING), the best model's prediction on ESR2 was a mediocre 0.627, even after training on 230k examples [9].  For this reason, docking scores have been proposed as promising benchmarks for drug discovery [9,10,11].
>
> We used two different datasets of docking scores: the original DOCKSTRING dataset, and an augmented dataset where scores were transformed either linearly or non-linearly (by taking the minimum or the maximum between each score and the median of the score distribution) and were combined with the quantitative estimate of drug-likelihood (QED).
>
> **References**
>
> [1] Finn et al 2017. Model-agnostic meta-learning for fast adaptation of deep networks. https://arxiv.org/abs/1703.03400
>
> [2] Garnelo et al 2018. Conditional neural processes. http://proceedings.mlr.press/v80/garnelo18a/garnelo18a.pdf
>
> [3] Garnelo et al 2018. Neural processes. https://arxiv.org/abs/1807.01622
>
> [4] Bruinsma et al 2023. Autoregressive Conditional Neural Processes. https://openreview.net/forum?id=OAsXFPBfTBh
>
> [5] Pat Walters.  We need better benchmarks for machine learning in drug discovery. http://practicalcheminformatics.blogspot.com/2023/08/we-need-better-benchmarks-for-machine.html
>
> [6] Chan et al 2023. Embracing assay heterogeneity with neural processes for markedly improved bioactivity predictions. https://arxiv.org/abs/2308.09086
>
> [7] Wong et al 2022. Benchmarking AlphaFold-enabled molecular docking predictions for antibiotic discovery https://www.embopress.org/doi/full/10.15252/msb.202211081
>
> [8] Gentile et al 2022. Artificial intelligence–enabled virtual screening of ultra-large chemical libraries with deep docking. https://www.nature.com/articles/s41596-021-00659-2
>
> [9] Garcia-Ortegon et al 2022. DOCKSTRING: Easy molecular docking yields better benchmarks for ligand design. https://pubs.acs.org/doi/full/10.1021/acs.jcim.1c01334
>
> [10] Huang et al 2021. Therapeutics Data Commons: Machine learning datasets and tasks for drug discovery and development. https://arxiv.org/abs/2102.09548
>
> [11] Ciepliński et al 2023. Generative models should at least be able to design molecules that dock well: A new benchmark. https://pubs.acs.org/doi/10.1021/acs.jcim.2c01355

---

### Official Review · Reviewer_P8e8 · 2023-11-03

**Soundness:** 3 good
**Presentation:** 4 excellent
**Contribution:** 2 fair
**Rating:** 5
**Confidence:** 4

**Summary:**

In this paper, the authors study Graph NP's performance in few-shot learning tasks, and propose fine-tuning strategies to further improve GNP's regression performance while maintaining good calibration. They also present a Bayesian optimization case study to showcase GNP's potential advantages.

**Strengths:**

1. Writing: Well-organized, easy-to-follow paper.
2. Significance: Show that graph NPs are competitive in molecular few-shot learning tasks.

**Weaknesses:**

1. Applicability: Focuses on regression tasks only despite abundant data and baselines in classification.
2. Novelty: Fine-tuning NPs during meta-testing are not novel contributions.

**Questions:**

1. Why didn't you extend Graph NPs into the classification setting, where the amount of data and baselines is abundant?
2. Could you explain the results presented in Figure F.1, where the $R^2$ did not decrease as the percentage of training points sampled increase?
3. Have you considered studying the impact of context/target set randomization on calibration of uncertainty estimates?

---

> ### Author Response · Authors · 2023-11-23
>
> We thank Reviewer P8e8 for their time. We address their comments below.
>
> **Could you explain the results presented in Figure F.1, where the did not decrease as the percentage of training points sampled increase?**
>
> This is indeed an interesting point. It appears that the NPs' mean prediction $\mu_\theta(x)$ was more robust to increasing the percentage of context points during meta-training than the variance prediction $\sigma_\theta^2(x)$. As a result, the NLPD deteriorated much more (Figure 4) than the $R^2$ (Figure F.1). This seems reasonable: if the model overfits to the training datapoints and memorizes their $y$ values, the mean predictions will fixate at the $y$ values and the variance predictions will become ever smaller (since smaller variances with the correct mean will lead to higher predictive likelihoods). Therefore, the model will learn to always predict inadequately small variances, which leads to catastrophic NLPD at meta-test times.
>
> It is worth noting, however, that even if the NLPD deteriorated faster than the $R^2$, the $R^2$ also suffered if the fraction of context points at each iteration during meta-training was too high. This is shown in Figure F1, where the $R^2$ value of test datapoints was always lower at 100% than at 0.8, 2, 4 or 8%, for every meta-train and every meta-test function.
>
>
> **Have you considered studying the impact of context/target set randomization on calibration of uncertainty estimates?**
>
> We thank the reviewer for making this suggestion. The NLPD in Figure 4 was partly a measure of calibration since poorly calibrated models have larger NLPD. Indeed, when 100% of points of a function were taken as contexts and targets (hence having no randomization of contexts and targets during meta-training) the NLPD increased significantly both for ftrain, dtrain points and for all meta-test points (ftest, dtrain and ftest, dtest). In addition, we have now added to the manuscript Figure F.2, which shows the correlation of predicted variance and prediction error in a MG-CNP sampling a small fraction of training points as contexts and targets at each iteration, as usual, and a MG-CNP taking 100% of points as contexts and targets. The latter displayed low correlation, indicating poor calibration.

---

### Author Response · Authors · 2023-11-23
**Response to all reviewers**

We sincerely thank the reviewers for their time and for their thoughtful suggestions, which have helped us improve the paper. In this general response we address some of the points raised by two reviewers. We reply to other points individually below.

**Lack of novelty of methods**

We understand this objection. However, we would like to emphasize that our paper is an applications paper, submitted to the area "Applications to physical sciences (physics, chemistry, biology, etc.)". The majority of the papers in the neural processes (NPs) literature have developed new methods, but very few have attempted to apply such methods to real problems. As a result, there exists a widening gap between the NP family, which has grown considerably in the last years, and their utility in real-world problems, which is still quite limited. In the specific case of drug discovery, they have only been applied to molecules very recently [1,2,3]. The goal of our paper is not to develop a mathematically-novel NP, but rather to take already-existing NPs and adapt them so that they can be applied effectively to a scientifically-interesting problem, namely meta-learning of (high-dimensional) drug-discovery datasets. In the process of doing so, we discovered the fragility of MAML and NPs, which fail catastrophically if meta-test functions are even slightly different from meta-train functions (Figure 1). For meta-learning to be useful in scientific applications, this problems needs to be addressed. Our proposed solution, parameter adaptation from the loss of the context predictions, is effective and simple to implement. As far as we are aware, our paper is the first to recognize that context predictions by NPs are not necessarily accurate, even if the NP has seen the true values of those context points, and that this can be exploited for parameter adaptation during meta-testing. We have benchmarked parameter adaptation extensively using multiple regression objectives, representations (fingerprints and molecular graphs, using a popular GNN architecture [4]), adaptation strategies (MAML and fine-tuning) and NP models (CNPs and LNPs). More generally, we have benchmarked NPs against baselines typical in chemoinformatics (single-task models on fingerprints, GNNs and pre-trained GNNs) and meta-learning (GNN with MAML, and now also ADKF-IFT). This type of detailed benchmarking is necessary in order to make successful use of NPs in scientific applications.

**Why not benchmarking on FS-Mol?**

We agree with the reviewers that FS-Mol is a popular dataset in molecular meta-learning. However, it presents an important limitation. Splitting molecular datasets is tricky because of the presence of chemical analogues, i.e. molecules that are structurally similar (often sharing the same scaffold) and display similar properties. Analogues are common because experiments are often carried out on chemical series of related molecules. Therefore, random splitting is not recommended because it may lead to data leakage and an overoptimistic estimation of performance. This fact is well known in the drug discovery literature, where random splitting is considered bad practice, especially in small datasets [5,6,7,8,9]. Unfortunately, FS-Mol only allows random splitting, which is why we did not employ it. We selected DOCKSTRING because it splits by scaffold, which makes prediction more challenging.

**Why not benchmarking on classification datasets?**

Biological experiments usually produce a real-number readout. Thereore, most classification drug-discovery datasets are real datasets that were binarized post hoc. For this reason, regression is more generally applicable to drug discovery. In addition, NPs are usually framed for regression, using a Gaussian likelihood. If we benchmarked NPs in a classification setting, using a Bernouilly likelihood, it is not clear that the resulting takeaways would also be applicable to a regression setting.

**Comparison with ADKF-IFT**

We thank the reviewers for suggesting this baseline, which is indeed highly relevant. We have now added ADKF-IFT [10] as a baseline, evaluating it in few-shot learning (Table 1), uncertainty calibration (Figure 2) and Bayesian optimization (Figure 5) (in updated PDF).

**References**

[1] Lee et al 2022.  https://openreview.net/forum?id=yzlif16IASM

[2] Chan et al 2023.  https://arxiv.org/abs/2308.09086

[3] Garcia-Ortegon et al 2022.  https://arxiv.org/abs/2210.09211

[4] Xiong et al 2020. https://pubs.acs.org/doi/10.1021/acs.jmedchem.9b00959

[5] Leonard and Roy 2006. https://onlinelibrary.wiley.com/doi/abs/10.1002/qsar.200510161

[6] Sheridan 2013. https://pubs.acs.org/doi/10.1021/ci400084k

[7] Martin et al 2017. https://pubs.acs.org/doi/10.1021/acs.jcim.7b00166

[8] Lopez-del-Rio et al 2019.  https://pubs.acs.org/doi/10.1021/acs.jcim.8b00663

[9] Puzyn et al 2011. https://link.springer.com/article/10.1007/s11224-011-9757-4#ref-CR9

[10] Chen et al 2022. https://arxiv.org/abs/2205.02708

---

### Meta-Review · Area_Chair_rYJo · 2023-12-05

**Metareview:**

- Claims and findings:

This submissions covers meta-learning on molecular tasks, leveraging Neural Processes (NP) to do so. In particular, the paper studies the application of graph NPs to drug discovery on DOCKSTRING which is a dataset of docking scores. The submission sets up experiments with an unusual meta-learning setting: the correlation between training and testing tasks are controlled at a low degree, and the size of context varies in a large range. To deal with this, authors propose to combine gradient-based adaptation (MAML fine-tuning) with the Graph NP model. Empirical results show that MG-CNPc(fine-tuned) has a performance advantage in most cases.

- Strengths:

Reviewers have pointed out that authors comprehensively study NP-based models on molecular tasks, including different NP variants, different molecular features, different additional adaptation strategies. In addition, data processing and empirical results are shown in detail. Reviewers, highlight that the experimental setting looks convincing and that the proposed method could show advantage in such setting.


- Weaknesses:

Reviewers have pointed out that a more effective way to evaluate the proposed approach would be to use Graph NPs in the classification setting, where the amount of data and baselines is abundant. Similarly, reviewers also pointed out that there are no empirical results of existing popular datasets (e.g., fs-mol[2], moleculenet[3]), nor comparing them with real-world cases. Reviewers have also noted that the related work section is poorly organized mixing several topics (i.e. datasets, methods) together, which are hard to read.

- Missing in submission:

If the authors had included empirical evaluation on fs-mol or moleculenet datasets I would be more leaning towards accepting the paper. As it currently stands it's hard for me to know if this is a valuable contribution or if its an approach the is particularly useful in the DOCKSTRING dataset.

**Justification For Why Not Higher Score:**

Relevant datasets are missing from the empirical evaluation.

**Justification For Why Not Lower Score:**

N/A

---

### Decision · Program_Chairs · 2024-01-16

Reject